# Negative elongation factor complex enables macrophage inflammatory responses by controlling anti-inflammatory gene expression

Li Yu[1,2,3,9], Bin Zhang [1,3,9], Dinesh Deochand [4], Maria A. Sacta[4,5], Maddalena Coppo[4], Yingli Shang[6], Ziyi Guo[1,3], Xiaomin Zeng[1,3], David A. Rollins[4,7], Bowranigan Tharmalingam[4], Rong Li[8], Yurii Chinenov[4], Inez Rogatsky [4,7,10 ✉] & Xiaoyu Hu [1,2,3,10 ✉]

Studies on macrophage gene expression have historically focused on events leading to RNA polymerase II recruitment and transcription initiation, whereas the contribution of post-initiation steps to macrophage activation remains poorly understood. Here, we report that widespread promoter-proximal RNA polymerase II pausing in resting macrophages is marked by co-localization of the negative elongation factor (NELF) complex and facilitated by PU.1. Upon inflammatory stimulation, over 60% of activated transcriptome is regulated by poly-merase pause-release and a transient genome-wide NELF dissociation from chromatin, unexpectedly, independent of CDK9, a presumed NELF kinase. Genetic disruption of NELF in macrophages enhanced transcription of AP-1-encoding *Fos* and *Jun* and, consequently, AP-1 targets including *Il10*. Augmented expression of IL-10, a critical anti-inflammatory cytokine, in turn, attenuated production of pro-inflammatory mediators and, ultimately, macrophage-mediated inflammation in vivo. Together, these findings establish a previously unappreciated role of NELF in constraining transcription of inflammation inhibitors thereby enabling inflammatory macrophage activation.

[1] Institute for Immunology and School of Medicine, Tsinghua University, Beijing, China. [2] Tsinghua-Peking Centre for Life Sciences, Tsinghua University, Beijing, China. [3] Beijing Key Laboratory for Immunological Research on Chronic Diseases, Beijing, China. [4] Hospital for Special Surgery Research Institute, The David Rosensweig Genomics Center, 535 East 70th Street, New York, NY, USA. [5] Weill Cornell/Sloan Kettering/Rockefeller Tri-Institutional MD-PhD Program, 1300 York Avenue, New York, NY, USA. [6] College of Veterinary Medicine, Shandong Agricultural University, Taian, China. [7] Graduate Program in Immunology and Microbial Pathogenesis, Weill Cornell Graduate School of Medical Sciences, 1300 York Avenue, New York, NY, USA. [8] Department of Biochemistry & Molecular Medicine School of Medicine & Health Sciences, The George Washington University, 2300 I Street NW, Washington, DC, USA. [9] These authors contributed equally: Li Yu, Bin Zhang. [10] These authors jointly supervised this work: Inez Rogatsky, Xiaoyu Hu. ✉email: rogatskyi@hss.edu; xiaoyuhu@tsinghua.edu.cn

nflammation has evolved as a rapid response to environmental cues and contributes to a series of physiological and pathological processes including host defense, tissue damage, and metabolic alterations[1–3]. The potentially harmful effects of inflammation necessitate the existence of precise regulatory mechanisms to control its magnitude and duration[4,5]. Indeed, activation of macrophages, an essential component of inflammatory responses, is subject to exquisitely tight control at multiple levels. For example, during activation, macrophage transcriptome undergoes extensive reprogramming with hundreds of genes rapidly upregulated and downregulated hundreds or even thousands of fold[6], pointing at transcription as a critical node of regulatory circuitry in these cells. Macrophage activation is typically triggered by ligation of cell surface or endosomal receptors which initiates intracellular signaling, culminating in recruitment of sequence-specific transcription factors and RNA polymerase (Pol) II to the target gene loci[7]. Historically, Pol II recruitment and transcription initiation were considered the major rate-limiting steps for gene activation[8]. However, this view was later challenged by several studies reporting that in resting macrophages, transcription start sites (TSS) of many inflammatory genes such as *Tnf* are preloaded by Pol II[9,10] raising the possibility that the rate-limiting steps to their activation occur post transcription initiation.

Indeed, numerous recent studies conducted mainly in *Drosophila* and stem cells have described Poll II promoter–proximal pausing, pause-release and entry into productive elongation as equally susceptible to regulation[11]. Specifically, after formation of the preinitiation complex (PIC), Pol II initiates transcription, synthesizes short (20–60 nt) nascent RNAs and then pauses. Further productive elongation requires signal-dependent pause-release to mobilize Pol II into the gene body regions. Given the importance of Pol II pausing, establishment of pause and its release are highly regulated by a plethora of positive and negative factors, including negative elongation factor (NELF), DRB sensitivity-inducing factor (DSIF), and positive transcription elongation factor-b (P-TEFb)[12,13]. In the canonical pause-release model derived from biochemical studies, the four-subunit NELF complex binds and retains Pol II within the promoter–proximal regions[14]. Pause-release is believed to be triggered by signal-induced phosphorylation of NELF by the heterodimeric P-TEFb complex composed of cyclin-dependent kinase 9 (CDK9) and cyclin T1, which results in dismissal of NELF from promoters. In addition, P-TEFb phosphorylates DSIF converting it from pausing to elongation-promoting factor and serine 2 residues within the heptad repeats in Pol II C-terminal domain (also targeted by CDK12), which together is thought to facilitate Pol II entry into gene bodies and productive transcription elongation[11,15].

Post-initiation regulation of transcription is implicated in key biologic processes, including embryogenesis and development[11,16–20]. The contribution of post-initiation mechanisms to immune cell function has not been widely appreciated although several pioneering studies have provided strong evidence for the existence of this type of regulation especially in cells such as macrophages that respond rapidly to environmental cues[9,10,21–23]. Ligation of TLR4 followed by NF−kB recruitment leads to P-TEFb binding to numerous gene loci[10,22,24,25]. In fact, studies by us and others have shown how P-TEFb loading and transcription elongation are targeted by negative regulators of inflammation including the glucocorticoid receptor and other transcription repressors[21,22,26], underscoring the physiological importance of immune gene regulation during early elongation. Nevertheless, these studies mainly focused on specific subsets of genes of interest, whereas the characteristics and a global impact of post-initiation control of transcription to macrophage activation remain to be thoroughly investigated.

Here, by employing genomic, pharmacological, and biochemical approaches, we comprehensively mapped the post-initiation transcriptional landscape during macrophage activation. We describe the surprisingly global and dynamic interactions of the "pausing factor" NELF with chromatin over the course of inflammatory activation of macrophages and the unexpected contribution of the lineage-determining transcription factor PU.1 to this process. Using genetic disruption of *Nelfb* in macrophages, we identify a functionally and transcriptionally diverse group of NELF-regulated genes that display aberrant responses to inflammatory signaling, and define a pathway linking paused genes under direct transcriptional control of NELF to their downstream effectors in the immune system. Finally, we describe the consequences of macrophage-specific NELF depletion in vivo thereby establishing a physiological role of NELF in mammalian inflammatory response.

## Results

**Widespread Pol II promoter–proximal pausing in macrophages.** To comprehensively define the global Pol II pausing patterns as related to signal-induced transcription in murine primary bone marrow-derived macrophages (BMDM), we performed Pol II chromatin immunoprecipitation followed by high throughput sequencing (chromatin immunoprecipitation (ChIP)-seq) and precision nuclear run-on sequencing (PRO-seq). Out of 10,076 unique genes expressed in BMDM as defined by RNA-seq (referred to as "BMDM transcriptome" hereafter), an overwhelming majority of genes displayed features of promoter–proximal pausing as computationally defined by high pausing index (PI) calculated based on Pol II ChIP-seq signals in the TSS regions versus gene body regions (Fig. 1a, b, Supplementary Fig. 1a). Highly paused (PI ≥ 3, group 1) and moderately paused (1.5 ≤ PI < 3, group 2) genes made up 76% of the BMDM transcriptome (Fig. 1c, Supplementary Fig. 1b), whereas non-paused genes made up 24% (PI < 1.5, group 3). The global Pol II pausing pattern was highly reproducible across independent ChIP-seq data sets (Supplementary Fig. 1c). To examine whether paused Pol II was transcriptionally active, we employed PRO-seq which detects de novo transcripts and found enriched promoter-proximal short transcripts in resting BMDM (Fig. 1d, e). Interestingly, PRO-seq based quantification also revealed promoter-proximal pausing in approximately 83% of the transcriptome (Supplementary Fig. 1d), which largely overlapped with Pol II pausing defined by ChIP-seq (Supplementary Fig. 1e). Therefore, both Pol II occupancy and production of nascent transcripts in resting BMDM corroborate promoter-proximal pausing as a widespread phenomenon that affects approximately three-quarters of BMDM transcriptome, consistent with Pol II pausing occurrence reported in other species and cell types[27].

To assess the role of factors associated with pausing and early elongation, we investigated the global occupancy of CDK9, the kinase subunit of P-TEFb, and the NELF− E subunit of the NELF complex in resting BMDM by ChIP-seq. P-TEFb plays a critical role in promoting elongation in a signal-dependent manner[28] and, consistently, showed little occupancy in resting BMDM (Fig. 1f, Supplementary Fig. 1f). In contrast, the "pausing factor" NELF displayed striking TSS-centric distribution with 71% of NELF-E ChIP-seq peaks located near TSS (Fig. 1g, h). NELF-E occupancy strongly correlated with paused Pol II and vice versa: highly paused genes showed the highest NELF-E binding signals (Fig. 1i, j), and 32% of group 1 genes displayed NELF-E peaks in their TSS regions (hereafter, NELF+ genes) compared to only 8% of group 3 (Supplementary Fig. 1g). Reciprocally, 75% of NELF+ genes were highly paused genes (Fig. 1k), as exemplified by *Tnf* and *Jun* (Fig. 1l). Thus, in resting BMDM, the majority of transcriptome displayed features of promoter-proximal pausing

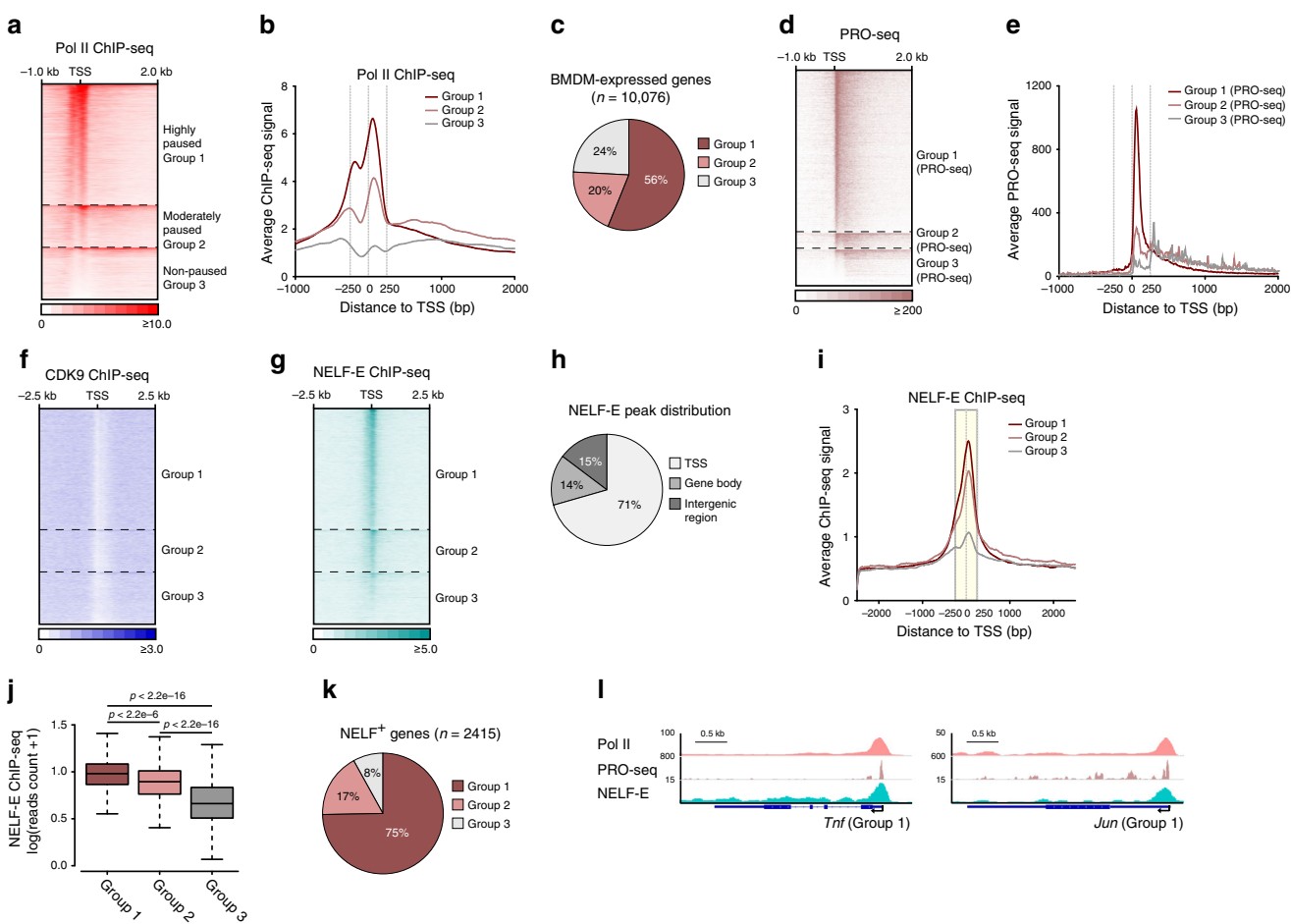

**Fig. 1 Pol II promoter–proximal pausing in resting BMDM is marked by NELF co-localization. a** Heat map of Pol II ChIP-seq signals around the TSS regions for BMDM-expressed highly paused (group 1), moderately paused (group 2), and non-paused (group 3) genes. Each row indicates one gene. In each group, the rows were sorted by the decreasing Pol II ChIP-seq signal in TSS region. **b** Average Pol II ChIP-seq signals around TSS regions of group 1 (brown), group 2 (red), and group 3 (gray) genes, as indicated. **c** Pie graph shows the percentage of BMDM-expressed genes in each group. **d** Heat map of PRO-seq signals (sense strand) around the TSS regions for three groups of genes classified based on PRO-seq defined PI. Each row indicates one gene. In each group, the rows were sorted as in (**a**). **e** Average PRO-seq signals (sense strand) around TSS regions of group 1–3 genes from (**d**). **f, g** Heat maps for CDK9 (**f**) and NELF-E (**g**) ChIP-seq signals around the TSS regions for three groups of genes. In each group, the rows were sorted as in (**a**). **h** The genomic distribution of NELF-E ChIP-seq peaks. TSS and gene body regions were defined as shown in Supplementary Fig. 1a. **i** Average NELF-E ChIP-seq signals around TSS of group 1–3 genes, as indicated. **j** NELF-E ChIP-seq reads quantified in −250 to +250 bp region (boxed in (**i**)) are shown as box-plots for group 1–3 genes, as indicated. Boxes outline the data from first quartile to third quartile, and bars indicate the boundary whose distribution does not exceed 1.5-fold of interquartile range extended from both first quartile and third quartile. $p$ Values (group 1/group 2 $P < 2.2e^{-16}$, group 1/group 3 $P < 2.2e^{-16}$, group 2/group 3 $P < 2.2e^{-16}$) were calculated by two-sided Mann–Whitney $U$ test. **k** The percentage of group 1–3 genes in NELF+ BMDM-expressed genes. **l** Tracks of Pol II ChIP-seq, PRO-seq (sense strand), and NELF-E ChIP-seq are shown for *Tnf* and *Jun* group 1 genes.

characterized by TSS-centric NELF occupancy co-localizing with that of Pol II, potentially rendering these genes primed to regulation at the post-initiation steps.

**PU.1 contributes to TSS-centric Pol II and NELF localization.** We next sought to identify factors that may contribute to establishing promoter–proximal pausing in macrophages. Motif analysis of DNA sequences in TSS regions of our three groups of genes revealed that ETS family transcription factor binding motifs were significantly enriched near TSS of highly paused genes (Fig. 2a; compare group 1 and 2-3). Out of 26 ETS family transcription factors, PU.1 is the lineage-determining factor and the highest expressed family member in macrophages (Fig. 2b). Interestingly, in addition to previously reported occupancy at gene enhancers[29], our analysis of published PU.1 ChIP-seq data[30] revealed that 58% of genes expressed in BMDM exhibited PU.1

occupancy near TSS (Fig. 2c). Moreover, TSS PU.1+ genes largely overlapped with the highly paused group 1 genes (Fig. 2d, e). PU.1 ChIP-seq signals showed significant positive correlation with the degree of pausing (Fig. 2f, g), with representative genes, e.g., *Tnf*, demonstrating concomitant Pol II, NELF-E, and PU.1 occupancy near TSS (Fig. 2h).

To probe for the causal direct or indirect relationship between PU.1 and Pol II promoter–proximal pausing, PU.1 expression was manipulated in immortalized macrophages using the CRISPR–Cas9 technique. Diminishing PU.1 expression in fully differentiated macrophages (Fig. 2i) attenuated both Pol II and NELF-E binding near the *Tnf* TSS (Fig. 2j). To further corroborate a positive role for PU.1 in TSS-centric Pol II accumulation, we mutated a PU.1 binding site near the endogenous *Tnf* TSS in immortalized macrophages using CRISPR–Cas9 technique (Supplementary Fig. 2a). ChIP experiments in the PU.1 binding site-mutated macrophages revealed

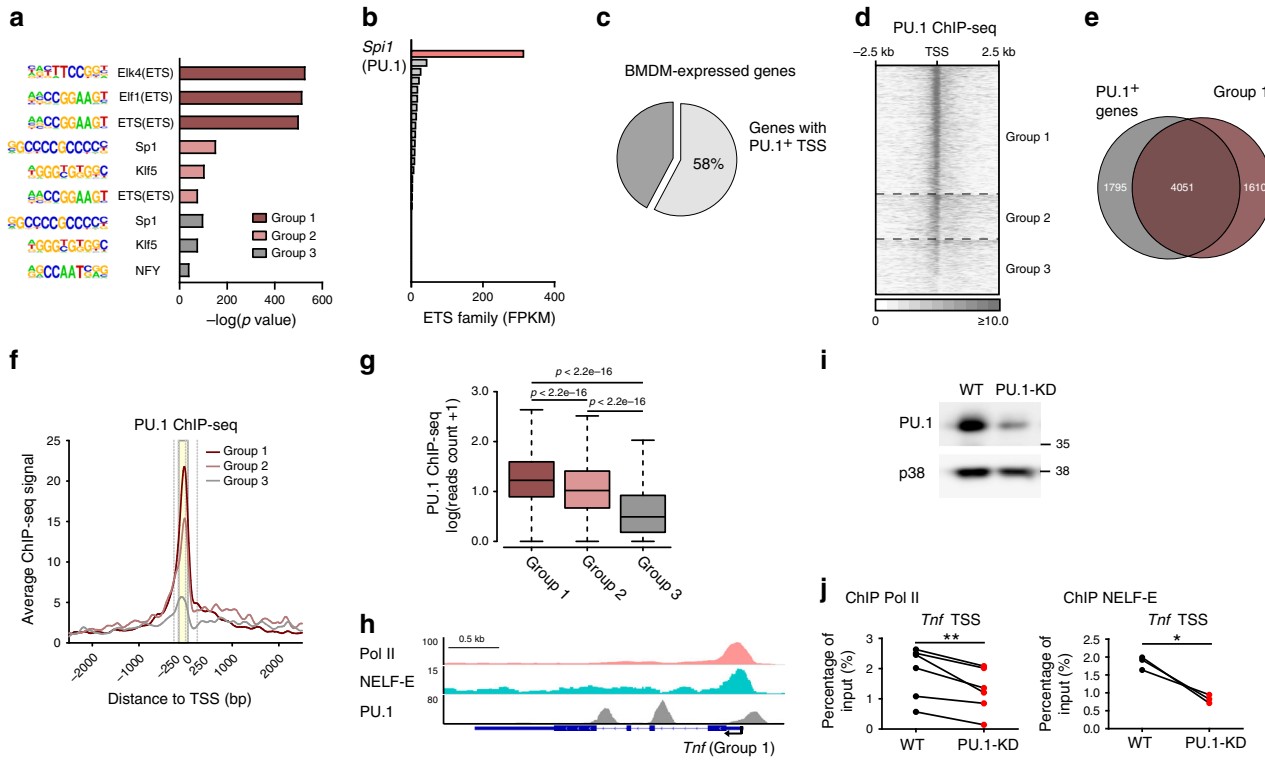

**Fig. 2 PU.1 contributes to establishment of promoter–proximal pausing in macrophages. a** Motif enrichment analysis in TSS regions of BMDM-expressed genes. Top three most enriched transcription factor binding motifs in TSS region of each gene group are shown; x-axis is $-\log_{10}(p$ value) for each enriched motif. Binomial distribution was used for $p$ value calculation, and no adjustments were made for multiple comparisons. **b** Expression level (fragments per kilobase of transcript per million mapped reads [FPKM]) measured by RNA-seq in resting BMDM for each of the 26 mouse ETS family transcription factors. **c** The percentage of genes with PU.1$^+$ TSS regions (PU.1$^+$ genes) in BMDM-expressed genes. **d** Heat map of PU.1 ChIP-seq signals in resting BMDM around the TSS regions for BMDM-expressed genes. For each group, the rows were sorted as in Fig. 1a. **e** The overlap between PU.1$^+$ BMDM-expressed genes and group 1 genes. **f** Average signals of PU.1 ChIP-seq in resting BMDM around TSS regions of group 1–3 genes, as indicated. **g** PU.1 ChIP-seq reads quantified in resting BMDM in $-150$ to $+50$ bp region (boxed in (**f**)) are shown as boxplots for group 1–3 genes, as indicated. The boxes and bars express data as in Fig. 1j. $p$ Values (group 1/group 2 $P < 2.2e^{-16}$, group 1/group 3 $P < 2.2e^{-16}$, group 2/group 3 $P < 2.2e^{-16}$) were calculated by two sided Mann–Whitney $U$ test. **h** Tracks of Pol II, NELF-E, and PU.1 ChIP-seq in resting BMDM are shown for *Tnf*, a representative group 1 gene. **i** Western blot for PU.1 and p38 (loading control) in WT and PU.1-KD iBMDM. Shown is one representative blot from three biological replicates. **j** The occupancy of Pol II and NELF-E in the TSS region of *Tnf* was assessed by ChIP-qPCR in WT and PU.1-KD iBMDM (Pol II: $n = 6$, $P = 0.0082$; NELF-E: $n = 3$, $P = 0.0277$). *$P < 0.05$, **$P < 0.01$ by two-sided paired Student's $t$ test. $n$ represents biologically independent experiments. Source data are provided as a Source Data file.

impaired Pol II and NELF-E occupancy near the TSS (Supplementary Fig. 2b–d), providing additional genetic evidence for the contribution of a PU.1 *cis*-element in the assembly of the pausing machinery.

**NELF dismissal is a key feature of macrophage activation**. Widespread Pol II pausing in resting BMDM suggested that pause-release may mediate activation of macrophage transcriptome by inflammatory signals. Following 1 h exposure of BMDM to LPS, a TLR4 agonist, activation of group 1 and 2 genes —approximately two-thirds of LPS-responsive genes—displayed a dramatic increase in both Pol II loading and elongation of paused Pol II into gene bodies indicative of pause-release (Fig. 3a, b, Supplementary Fig. 3a). In contrast, only 32% of LPS-induced macrophage transcriptome were activated solely by de novo recruitment of Pol II to promoters (Fig. 3a, b, group 3 genes). Pause-release in response to LPS was highly reproducible (Supplementary Fig. 3b) and was mirrored by a significant decrease in PI of group 1 as well as group 2 genes in the LPS-activated compared to the resting BMDM (Fig. 3c, Supplementary Fig. 3c). To determine whether Pol II released into gene body regions was transcriptionally active, we assessed de novo transcription by PRO-seq. As expected, PRO-seq captured

transcriptional pausing in over half (71%) of LPS-induced genes in resting BMDM (Supplementary Fig. 3d), as well as increasing transcriptional events across gene bodies and significantly attenuated PRO-seq-defined PI upon LPS treatment, further reinforcing the notion that pause-release acts as a hallmark of activation of group 1 genes (Fig. 3d–f). Representative genes from group 1 (*Tnf*) and group 3 (*Il1b*) along with their PI are shown in Fig. 3g.

Next, we sought to identify key factors that mediated differential regulation of paused vs. non-paused genes. P-TEFb is a positive regulator of transcription elongation, implicated in gene activation[11,12,31]. Measuring global CDK9 occupancy in LPS-activated BMDM by ChIP-seq (Supplementary Fig. 3e) revealed similar CDK9 recruitment patterns in group 1 and group 3 genes (Supplementary Fig. 3f). Indeed, enhanced CDK9 recruitment in LPS-activated BMDM correlated with increased Pol II traveling into gene body regardless of the gene pausing status (Supplementary Fig. 3g). Because CDK9 recruitment was not a feature unique to paused genes, we next assessed the behavior of the pausing factor NELF during macrophage activation. Among LPS-activated genes, NELF-E occupancy correlated well with the pausing status as illustrated by prevalent NELF positivity in group 1 genes and scarcity of NELF in group 3

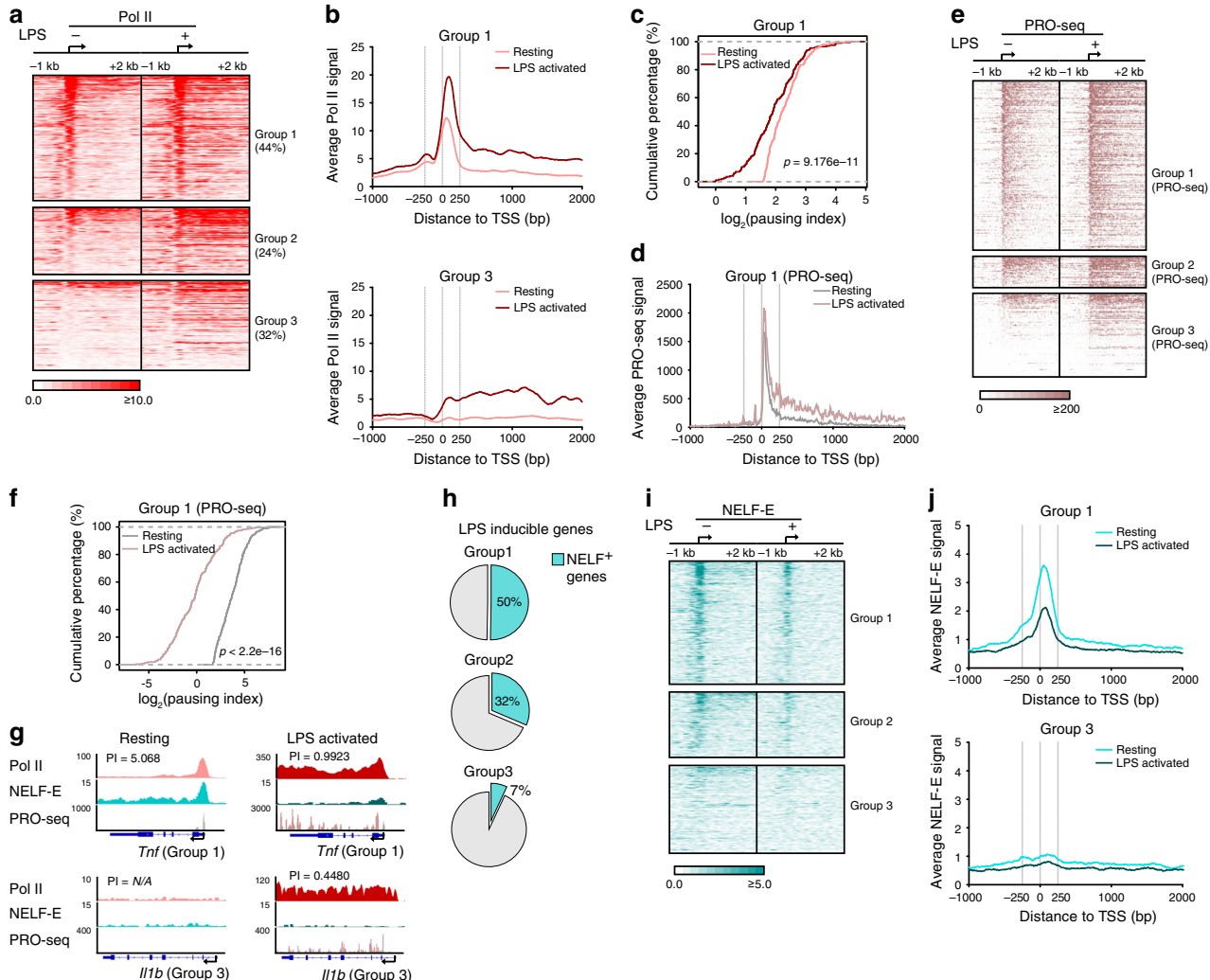

**Fig. 3 Macrophage activation is associated with Pol II pause-release. a** Heat map of Pol II ChIP-seq signals in resting and LPS-activated (1 h) BMDM around the TSS regions of group 1–3 LPS-inducible genes. For each group, the rows were sorted by the decreasing Pol II ChIP-seq signal in the TSS region in resting BMDM. **b** Average Pol II ChIP-seq signals in resting and LPS-activated BMDM around TSS of group 1 (top) and group 3 (bottom) LPS-inducible genes. (**c**) The empirical cumulative distribution function (ECDF) plot of Pol II PI distribution of group 1 LPS-inducible genes in resting and LPS-activated BMDM. $p$ Value (resting/LPS activated $p = 9.176e^{-11}$) was calculated by two-sided Kolmogorov–Smirnov test. **d** Average PRO-seq signals (sense strand) in resting and LPS-activated BMDM around TSS of PRO-seq group 1 LPS-inducible genes. **e** Heat map of PRO-seq signals (sense strand) in resting and LPS-activated (0.5 h) BMDM around the TSS regions of group 1–3 LPS-inducible genes that were classified by PRO-seq defined PI. For each group, the rows were sorted by decreasing PRO-seq signals (sense strand) in TSS regions. **f** The empirical cumulative distribution function (ECDF) plot of PRO-seq-defined PI of group 1 LPS-inducible genes in resting and LPS-activated (0.5 h) BMDM. $p$ Value (resting/LPS activated $P < 2.2e^{-16}$) was calculated by two-sided Kolmogorov–Smirnov test. **g** Tracks of Pol II ChIP-seq, NELF-E ChIP-seq, and PRO-seq (sense strand), as indicated, are shown for representative group 1 (*Tnf*) and group 3 (*Il1b*) genes in resting (left) and LPS-activated (right) BMDM. **h** The percentage of NELF⁺ genes in group 1–3 of LPS-inducible genes. **i** Heat map of NELF-E ChIP-seq signals in resting and LPS-activated (0.5 h) BMDM around the TSS regions of group 1–3 LPS-inducible genes. For each group, the rows were sorted as in (**a**). **j** Average NELF-E ChIP-seq signals in resting and LPS-activated BMDM around TSS of group 1 (top) and group 3 (bottom) LPS-inducible genes.

genes (Fig. 3h, Supplementary Fig. 3h). Interestingly, across multiple NELF-E ChIP-seq data sets, NELF occupancy dramatically declined within 0.5 h of LPS stimulation (Fig. 3i, j, Supplementary Fig. 3i, j). These results indicate that Pol II pause-release and NELF dissociation were hallmark events in inflammatory activation of macrophage transcriptome.

**NELF dissociation from chromatin is global yet transient.** Given a dramatic NELF loss from promoters in response to LPS (Supplementary Fig. 4a), we wondered whether such dismissal was restricted to LPS-inducible genes or occurred at a global scale across genome. Interestingly, NELF-E ChIP-seq revealed a highly dynamic pattern of NELF-E occupancy, in which NELF complex was globally dismissed from thousands of NELF⁺ genes 0.5 h post LPS stimulation (Fig. 4a, b, Supplementary Fig. 4b, c; approximately half of NELF⁺ genes), and reloaded to the TSS regions by 1 h approaching levels seen in resting BMDM (Fig. 4a, b, Supplementary Fig. 4b, d, e). Indeed, NELF dissociation was not restricted to LPS-inducible genes as illustrated by striking loss of NELF despite lack of transcriptional response from two representative paused genes *Ldha* and *Irf8* (Supplementary Fig. 4f). ChIP-seq data was corroborated biochemically, whereby the abundance of chromatin-associated NELF (assessed by immunoblotting for NELF-E and NELF-B subunits) decreased dramatically after 0.5 h of LPS exposure and was fully restored by 1 h

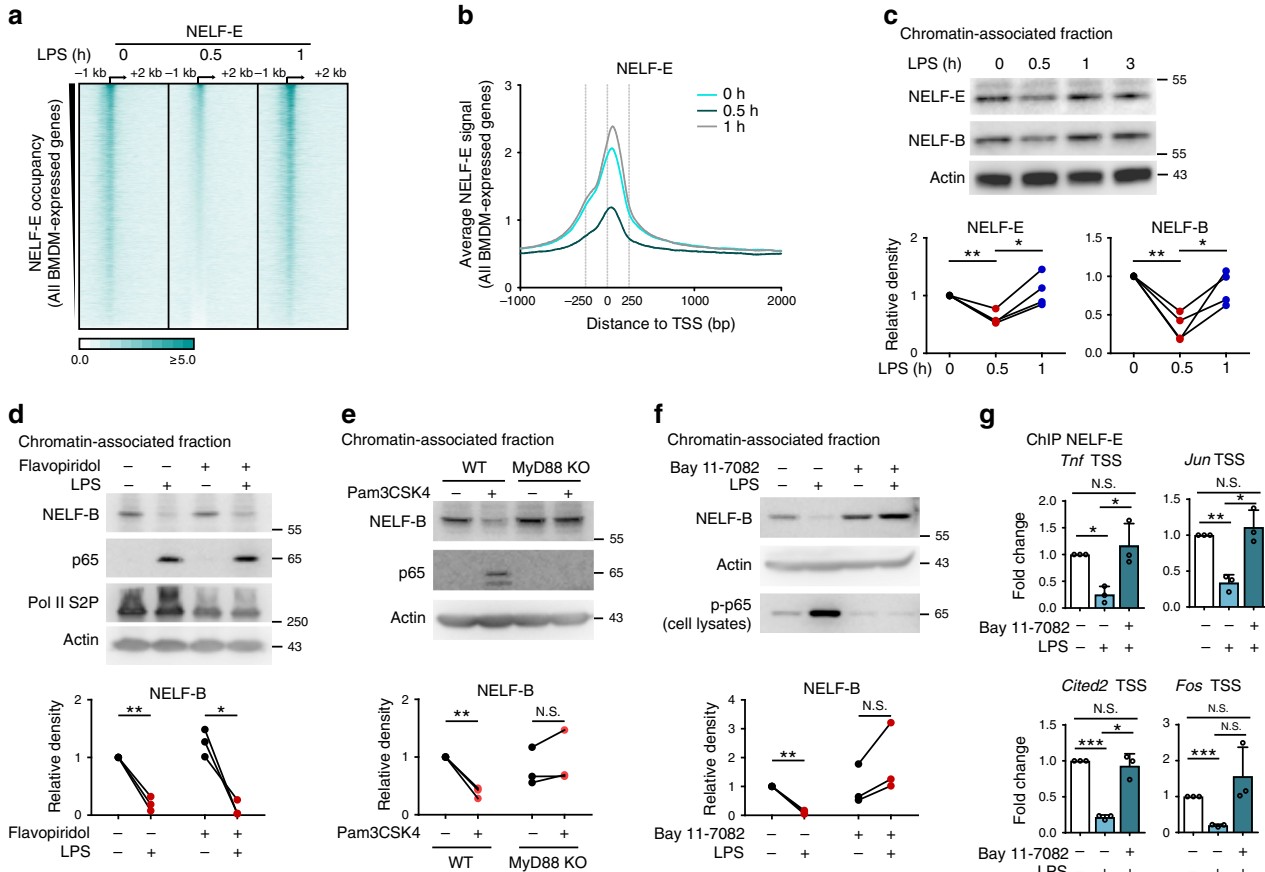

**Fig. 4 NELF is globally and transiently released from chromatin upon macrophage activation. a**, **b** Heat map (**a**) and average signals (**b**) of NELF-E ChIP-seq around TSS of all BMDM-expressed genes ($n = 10076$ genes) for BMDM treated with LPS as indicated. The rows in **a** were sorted by decreasing NELF-E occupancy in TSS regions in resting BMDM. **c** Immunoblotting of chromatin-associated NELF-E, NELF-B, and actin in BMDM treated as indicated ($n = 4$). NELF-E and NELF-B bands were quantified by densitometry, normalized to internal control (actin), and expressed relative to untreated (0 h) sample (=1) (bottom) (NELF-E LPS 0 h/0.5 h $P = 0.0063$, LPS 0.5 h/1 h $P = 0.0138$; NELF-B LPS 0 h/0.5 h $P = 0.005$, and LPS 0.5 h/1 h $P = 0.0314$). **d** Immunoblotting of chromatin-associated NELF-B, p65, Pol II S2P, and actin in mock (PBS) and flavopiridol (300 nM)-pretreated (0.5 h) BMDM followed by LPS (0.5 h) ($n = 3$). NELF-B bands in (**d**, **e**, **f**) were quantified and normalized as in (**c**) (bottom) (LPS 0 h/0.5 h $P = 0.0077$, flavopiridol pretreated LPS 0 h/0.5 h $P = 0.0322$). **e** Immunoblotting of chromatin-associated NELF-B, p65, and actin in WT and MyD88 KO BMDM with or without Pam3CSK4 stimulation (10 ng/ml for 0.5 h) ($n = 3$). Statistics of NELF-B bands (bottom) (WT/WT + Pam3csk4, $P = 0.0081$; MyD88 KO/MyD88 KO + Pam3csk4, $P = 0.2311$). **f** Immunoblotting of chromatin-associated NELF-B and actin in mock (DMSO) and Bay 11–7082 (5 μM)-pretreated (0.5 h) BMDM followed by LPS (0.5 h). p-p65 blot is control for Bay 11-7082 activity ($n = 3$). Statistics of NELF-B bands (bottom) (LPS 0 h/0.5 h, $P = 0.0016$; Bay 11-7082 pretreated LPS 0 h/0.5 h, $P = 0.1042$). **g** NELF-E occupancy in *Tnf*, *Jun*, *Cited2*, *Fos* TSS regions was assessed by ChIP-qPCR in mock (DMSO) and Bay 11–7082 (5 μM)-pretreated (0.5 h) BMDM followed by LPS (0.5 h) ($n = 3$) (*P* values for different comparative conditions in *Tnf*, *Jun*, *Cited2*, and Fos TSS are listed respectively: LPS 0 h/0.5 h $P = 0.0137, 0.0094, 0.0006, 0.0006$; LPS 0.5 h/LPS 0.5 h+Bay 11–7082, $P = 0.0345, 0.0245, 0.023, 0.1082$; LPS 0 h/LPS 0.5 h+Bay 11–7082, $P = 0.5634, 0.5074, 0.5464, 0.3562$). Mean + SD. *$P < 0.05$, **$P < 0.01$, ***$P < 0.001$, N.S. $P > 0.05$ by two-sided paired Student's *t* test. *n* represents biologically independent experiments. Source data are provided as a Source Data file.

(Fig. 4c). Of note, LPS-induced NELF chromatin dismissal and reloading were not a consequence of alterations in total cellular levels of the NELF complex as NELF-B and NELF-E protein abundance remained largely unchanged during the course of macrophage activation (Supplementary Fig. 4g).

In the canonical pause-release model, P-TEFb mediates NELF dismissal from the paused Pol II[32]. We assessed whether such model applied to dynamic changes of NELF occupancy during macrophage activation. Upon treatment of BMDM with flavopiridol, a CDK9 inhibitor, prior to and throughout LPS exposure, NELF still markedly dissociated from chromatin in response to LPS (Fig. 4d). Moreover, NELF was similarly dismissed from chromatin in immortalized macrophages in which CDK9 was depleted via CRISPR/Cas9-mediated gene editing (Supplementary Fig. 4h, i), providing genetic evidence for the P-TEFb-independent NELF release in macrophages.

We next confirmed that the integrity of the TLR pathway leading to NF-κB activation was required for NELF dissociation. Because TLR4 activates MyD88 and, to lesser extent, TRIF adapters, we used Pam3CSK4, a TLR1/2 agonist to activate specifically MyD88. As expected, Pam3CSK4 induced NELF dismissal from chromatin in BMDM, and MyD88 deletion abrogated this release (Fig. 4e). Furthermore, Bay 11-7082, an I-kappa-kinase (IKK) inhibitor—but not SB203580, an inhibitor of p38 MAPK less relevant to NF-κB activation—also prevented LPS-induced NELF chromatin dismissal (Fig. 4f, Supplementary Fig. 4j). The reversal of NELF release by IKK inhibition was apparent at individual target genes, e.g., *Tnf*, *Jun*, *Cited2*, *Fos* TSS regions as demonstrated by ChIP-quantitative polymerase chain reaction (qPCR) (Fig. 4g). Thus, activation of macrophage transcriptome by LPS led to a transient MyD88-IKK-dependent yet CDK9-independent NELF dismissal from chromatin.

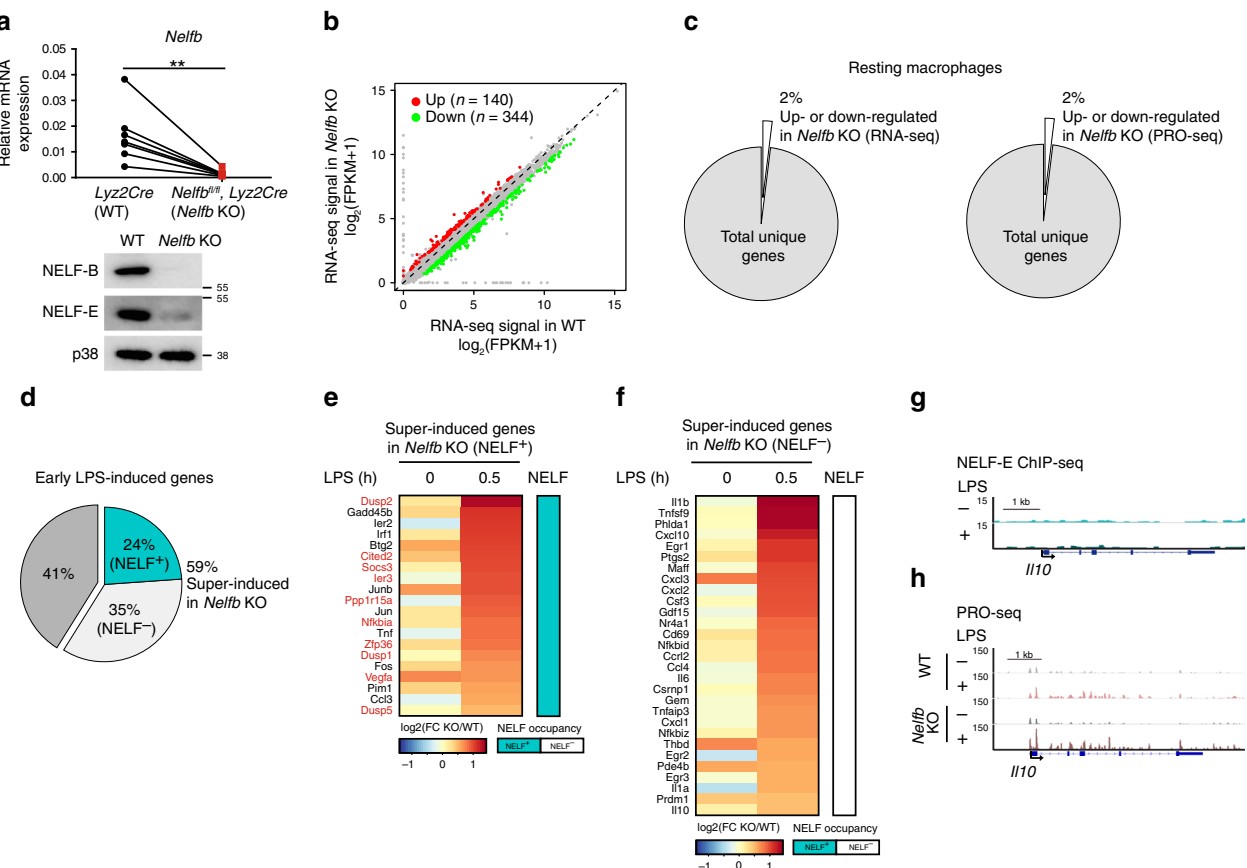

**Fig. 5 NELF regulates early inducible transcription activation in macrophages. a** Assessment of *Nelfb* KO efficiency. *Nelfb* mRNA (top) was measured by RT-qPCR in BMDM from multiple *Lyz2Cre* (WT) and *Nelfb*^fl/fl, *Lyz2Cre* (*Nelfb* KO) mice ($n = 7$) ($P = 0.0066$); **$P < 0.01$ by two-sided paired Student's *t* test. Western blot (bottom) analysis for NELF-B, NELF-E, and p38 in resting BMDM from WT and *Nelfb* KO mice. **b** Differentially expressed genes in resting macrophages assessed by RNA-seq. *x*- and *y*-axes correspond to normalized RNA-seq counts (FPKM) for total unique genes in WT and *Nelfb* KO BMDM, respectively. Upregulated and downregulated genes are highlighted in red and green, respectively. **c** The percentage of up- or down-regulated genes assessed by RNA-seq (left) and PRO-seq (right) in resting *Nelfb* KO BMDM compared to WT among total unique genes. **d** The percentage of genes super-induced in *Nelfb* KO BMDM among early (0.5 h) LPS-induced genes ($n = 83$). **e, f** Heat maps show the RNA-seq expression level (FPKM + 1) fold changes (*Nelfb* KO/WT) of NELF^+ (**e**) and NELF^− (**f**) genes super-induced in *Nelfb* KO in resting and LPS-treated (0.5 h) BMDM. Anti-inflammatory genes in (**e**) are shown in red. **g** NELF-E ChIP-seq tracks are shown for the *Il10* locus in resting (−) and LPS-activated (0.5 h) (+) BMDM. **h** PRO-seq (sense strand) tracks for the *Il10* locus in WT and *Nelfb* KO are shown for resting (−) or LPS-stimulated (0.5 h) (+) BMDM. Source data are provided as a Source Data file.

**NELF regulates early inducible transcription activation**. To investigate the role of NELF in macrophage function, we took a genetic approach by generating mice with myeloid cell-specific deletion of the *Nelfb* gene (*Nelfb*^fl/fl, *Lyz2Cre*, referred to as *Nelfb* KO hereinafter[22]). Consistent with previous reports in other cell types[33], deletion of *Nelfb* led to destabilization of the NELF complex and degradation of other subunits including NELF-E (Fig. 5a), confirming it to be an informative loss-of-function model for studying the role of the NELF complex in macrophages. Depletion of NELF in myeloid cells resulted in no gross abnormalities of animals or apparent alterations of macrophage populations in vivo under homeostatic condition (Supplementary Fig. 5a–c). Moreover, RNA-seq and PRO-seq showed that NELF deficiency did not significantly or systematically alter transcriptomes of resting BMDM with <2% of all expressed genes affected (Fig. 5b, c). Thus, NELF was largely dispensable for maintaining baseline macrophage gene expression, in line with grossly normal phenotypes of *Nelfb* KO mice.

Next, we examined the role of NELF during macrophage activation by exposing *Nelfb* KO BMDM to LPS. Strikingly, 49 of

83 genes induced by 0.5 h of LPS stimulation (59%) were superinduced in *Nelfb* KO compared to WT controls pointing at a profound role of NELF in early inducible transcription (Fig. 5d). Interestingly, among 161 genes whose expression was suppressed by LPS, NELF-E ChIP-seq revealed similar LPS-induced NELF dismissal from group 1 and 2 genes (Supplementary Fig. 5d) as seen for LPS-induced genes. However, NELF deficiency did not profoundly alter expression levels of LPS-downregulated genes (Supplementary Fig. 5e). Intriguingly, among genes superinduced by LPS in *Nelfb* KO, NELF^+ and NELF^− genes were almost evenly represented (Fig. 5d–f), suggesting both direct and indirect mechanisms for their regulation. Among paused NELF^+ LPS-super-induced genes, we noted a striking overrepresentation of inflammation inhibitors (highlighted in red in Fig. 5e). For example, Cbp/p300-interacting transactivator (Cited) 2 blocks NF-κB interaction with histone acetyltransferase p300[34]. We confirmed upregulated steady state mRNA as well as de novo transcripts of a representative NELF^+ anti-inflammatory gene *Cited2* in *Nelfb* KO by qPCR and PRO-seq, respectively (Supplementary Fig. 5f, g). Conversely, the *Il10* gene encoding a

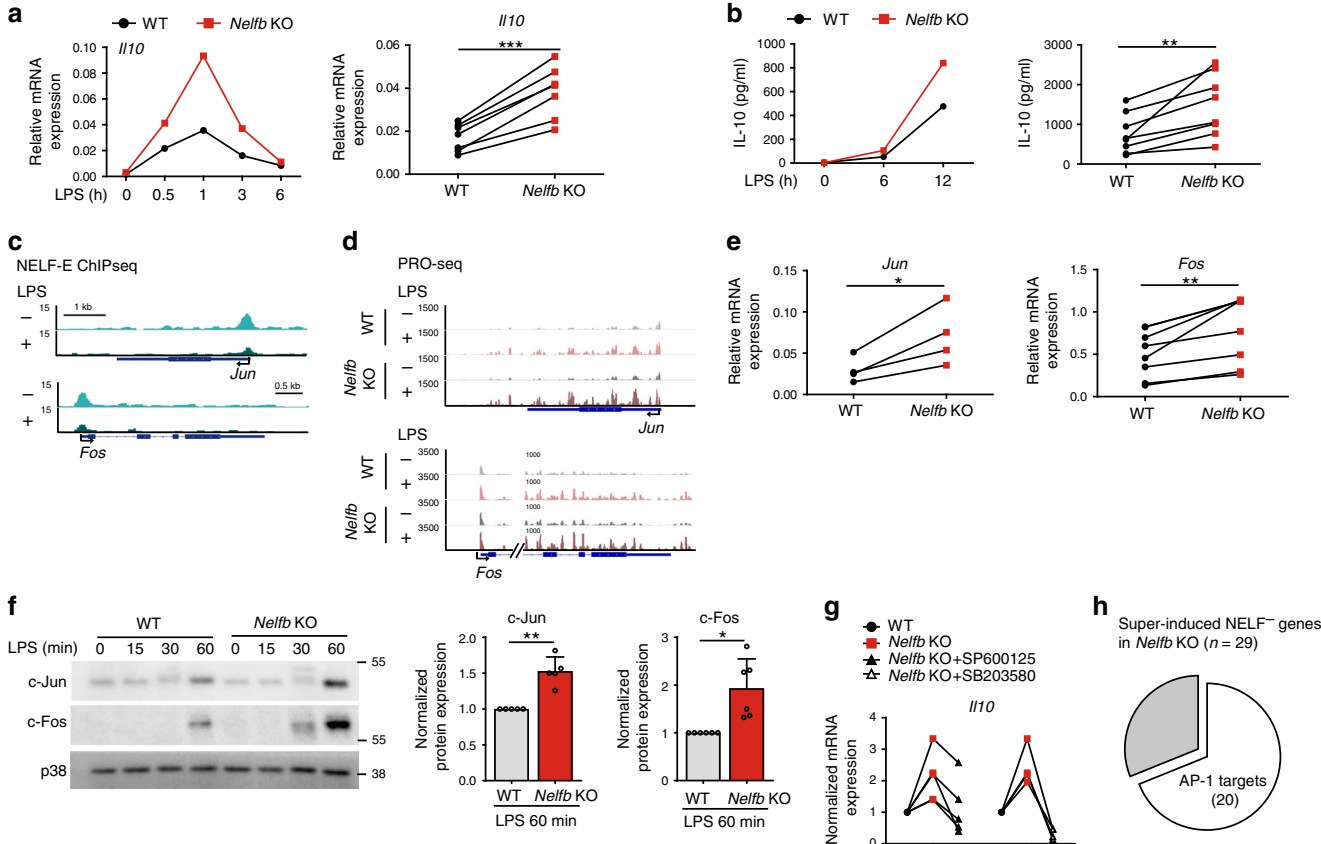

**Fig. 6 NELF controls IL-10 expression by constraining the transcription of AP-1. a** RT-qPCR analysis of *Il10* mRNA in WT and *Nelfb* KO BMDM stimulated with LPS for indicated time. Representative (left) and cumulative (LPS 0.5 h, n = 7, P = 0.0002) (right) data are shown. **b** IL-10 protein ELISA in WT and *Nelfb* KO BMDM stimulated with LPS for indicated time. Representative (left) and cumulative (LPS 6 h, n = 8, P = 0.0066) (right) data are shown. **c** NELF-E ChIP-seq tracks *Jun* and *Fos* in resting (−) and LPS-activated (+) (0.5 h) BMDM. **d** PRO-seq (sense strand) tracks for *Jun* and *Fos* in WT and *Nelfb* KO BMDM cultured with (+) or without (−) LPS for 0.5 h. **e** Cumulative RT-qPCR data showing expression level of *Jun* (n = 4, P = 0.0301) and *Fos* (n = 6, P = 0.0039) in WT and *Nelfb* KO BMDM treated with LPS for 0.5 h. **f** Immunoblot analysis of c-Fos, c-Jun, and p38 in whole cell lysates of WT and *Nelfb* KO BMDM treated with LPS for indicated time (left: a representative result of five replicates). For each replicate, c-Fos and c-Jun bands were quantified by densitometry at 60 min time point, normalized to internal control (p38) and expressed relative to WT ( = 1) (c-Jun P = 0.0041; c-Fos P = 0.0144). Mean + SD. **g** WT and *Nelfb* KO BMDM were pretreated with SP600125 (50 μM) or SB203580 (20 μM) for 0.5 h, where indicated, and *Il10* expression following 0.5 h of LPS co-incubation was assessed by RT-qPCR (n = 5). Relative *Il10* expression was normalized to the levels in WT cells that were set as 1. **h** Out of 29, 20 NELF⁻ genes super-induced in *Nelfb* KO are potential (P < 0.001) AP-1 targets. *P < 0.05, **P < 0.01, ***P < 0.001, N.S. P > 0.05 by two-sided paired Student's t test. n represents biologically independent experiments. Source data are provided as a Source Data file.

broad-spectrum anti-inflammatory cytokine was devoid of NELF binding yet, markedly transcriptionally super-induced by LPS in NELF-deficient BMDM (Fig. 5g, h), consistent with secondary effects of NELF on *Il10*, thus underscoring a multifaceted nature of the inhibitory impact of NELF on inflammatory transcriptome of macrophages.

**NELF controls an AP-1-dependent circuit to target IL-10.** Given the importance of IL-10 in immune regulation, we focused on the *Il10* gene as a model for understanding the indirect effects of NELF on macrophage transcriptome. First, we confirmed the RNA-seq and PRO-seq results by RT-qPCR in *Nelfb* KO BMDM, which showed heightened *Il10* transcript induction peaking by 1 h of LPS exposure (Fig. 6a, Supplementary Fig. 6a). Second, gene expression data was corroborated by ELISA that revealed over-production of IL-10 protein in *Nelfb* KO BMDM (Fig. 6b). Given that *Il10* is a NELF⁻ gene, we hypothesized that NELF inhibited IL-10 production by targeting factor(s) that promoted *Il10* expression. Among NELF-repressed genes (Fig. 5e), *Jun* and *Fos* encoding the subunits of the heterodimeric AP-1 transcription factor complex, are both direct NELF targets with prominent

LPS-sensitive NELF-E occupancy around their TSS regions (Fig. 6c). NELF deletion in BMDM resulted in enhanced de novo transcription of *Jun* and *Fos* genes as shown by PRO-seq (Fig. 6d). Consistently, Fos and Jun were upregulated in *Nelfb* KO BMDM at the mRNA and protein levels (Fig. 6e, f, Supplementary Fig. 6b), strongly implicating NELF as a negative regulator of AP-1.

AP-1 is an essential transcription factor for driving expression of a plethora of immune mediators including IL-10[35], making it plausible that NELF controls IL-10 expression by inhibiting transcription of AP-1. Indeed, pharmacological inhibition of AP-1 upstream signaling modules JNK and p38 MAPKs by SP600125 and SB203580, respectively, abrogated hyper-induction of *Il10* in *Nelfb* KO (Fig. 6g). Conversely, NELF deficiency did not alter TLR-induced activation of canonical NF-κB and MAPK signaling events (Supplementary Fig. 6c). To assess whether attenuation of AP-1 transcription by NELF was potentially affecting other NELF⁻ genes undergoing super-induction by LPS in *Nelfb* KO (Fig. 5f), we analyzed them for consensus transcription factor binding motifs within −1000 bp to +100 bp relative to TSS[36]. Interestingly, 20 of these genes (n = 29) were in fact AP-1 targets (Fig. 6h) and LPS increased Fos and Jun occupancy at *Il1b*, *Il10*,

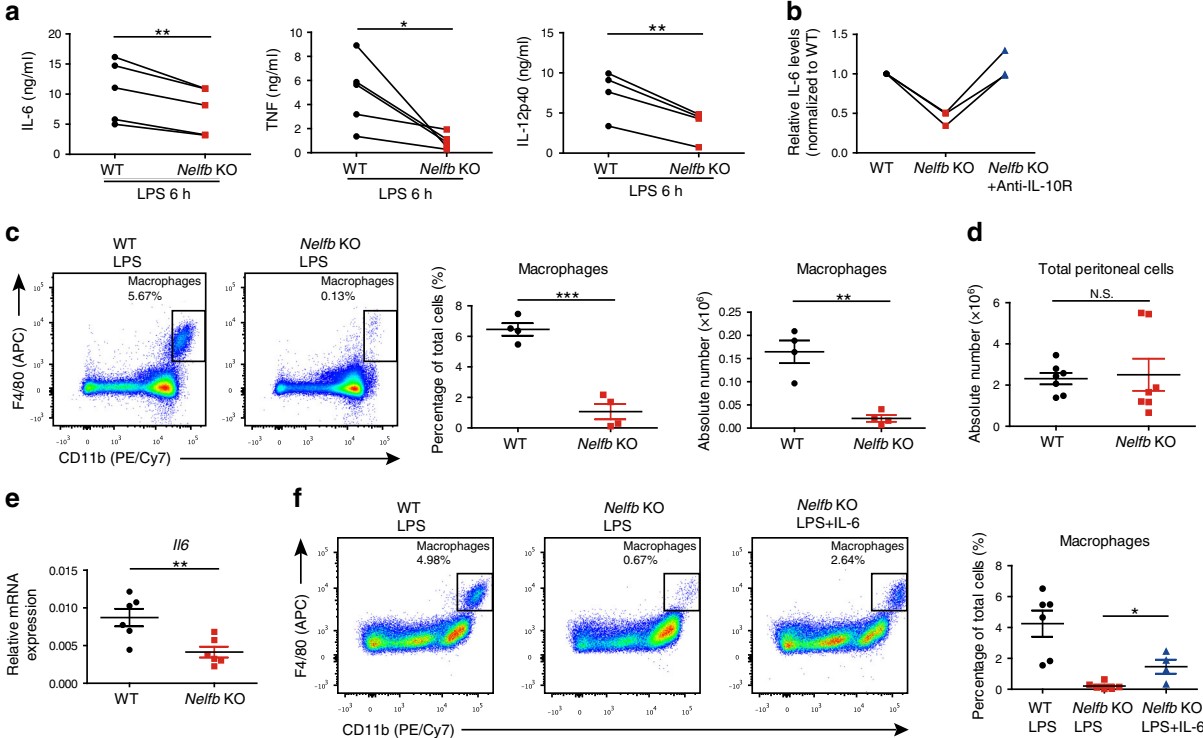

**Fig. 7 NELF potentiates inflammatory responses secondary to constraining the production of IL-10. a** ELISA for IL-6 ($n = 5$, $P = 0.0051$), TNF ($n = 5$, $P = 0.0402$) and IL-12p40 ($n = 4$, $P = 0.0061$) protein levels in WT and *Nelfb* KO BMDM stimulated with LPS for 6 h. *$P < 0.05$, **$P < 0.01$ by two-sided paired Student's $t$ test. **b** WT and *Nelfb* KO BMDM pretreated with 2 μg/ml of anti-IL10R for 0.5 h, where indicated, were activated with LPS for 6 h. IL-6 levels in culture supernatants were measured by ELISA and shown as relative values normalized to levels in WT cells that were set as 1 in each independent experiment ($n = 3$). **c** FACS analysis of the macrophage population (CD45+CD11b+F4/80+) in peritoneal exudates of WT and *Nelfb* KO mice 9 h after intraperitoneal injection of LPS (100 ng/mouse). Representative FACS distribution (left) and cumulative ($n = 4$) percentage (middle, $P = 0.0002$) as well as absolute numbers (right, $P = 0.0014$) are shown. **d** Cumulative data showing total peritoneal cell numbers from WT and *Nelfb* KO mice 9 h after LPS injection ($n = 7$, $P = 0.8232$). **e** RT-qPCR analysis of *Il6* in peritoneal cells of WT and *Nelfb* KO mice 9 h after intraperitoneal injection of LPS ($n = 6$, $P = 0.0074$). **$P < 0.01$ by two-sided paired Student's $t$ test. **f** FACS analysis of macrophage population (CD45+CD11b+F4/80+) in peritoneal exudates of WT and *Nelfb* KO mice 9 h after intraperitoneal injection of LPS or LPS + IL-6 (40 ng/mouse). Representative FACS distribution (left) and percentage of macrophages from independent experiments ($n = 4$–6) (right, WT + LPS/*Nelfb* KO + LPS $P = 0.0008$; *Nelfb* KO + LPS/*Nelfb* KO + LPS + IL-6 $P = 0.0105$) are shown. For **c**, **d**, and **f**, *$P < 0.05$, **$P < 0.01$, ***$P < 0.001$, N.S. $P > 0.05$ by two-sided unpaired Student's $t$ test. $n$ represents biologically independent animals. Error bars indicate SEM. Source data are provided as a Source Data file.

and *Tnfaip3* gene promoters (Supplementary Fig. 6d, e), consistent with the notion that NELF regulated a subset of activated macrophage transcriptome by controlling AP-1. Among such indirect targets of NELF downstream of AP-1 were a number of genes encoding both anti- and pro-inflammatory mediators including *Tnfaip3* and *Il1b* (Supplementary Fig. 6f). Taken together, NELF controlled expression of key immune regulators such as IL-10 by targeting an AP-1-dependent transcriptional circuit.

**NELF positively regulates macrophage-mediated inflammation.** As our results implied that NELF controlled the expression of important immune mediators including IL-10, we sought to assess the general contribution of NELF to inflammatory cytokine production by macrophages in vitro and in vivo. Corroborating a positive role of NELF in inflammation, its deficiency attenuated the production of IL-6, TNF, and IL-12p40 in response to LPS in cultured BMDM (Fig. 7a). Moreover, blocking the biological activities of IL-10 in *Nelfb* KO BMDM with anti-IL-10R antibody completely rescued IL-6 production (Fig. 7b) suggesting that reduced cytokine expression in *Nelfb* KO BMDM is at least in part due to augmented levels of IL-10.

To investigate the physiological role of NELF in controlling inflammatory cytokine balance in vivo, we employed the acute

sterile peritonitis model in which recruitment of inflammatory macrophages is driven by IL-6[37]. By examining peritoneal cell populations in this model (Supplementary Fig. 7a), we found that NELF deficiency in the myeloid lineage dramatically compromised infiltration of F4/80+CD11b+ macrophages into the peritoneal cavity (Fig. 7c) without affecting total number of peritoneal cells or the fraction of other peritoneal myeloid cell populations such as monocytes and neutrophils (Fig. 7d, Supplementary Fig. 7b, c). Moreover, peritoneal cells in *Nelfb* KO mice contained less *Il6* mRNA compared to WT controls (Fig. 7e), suggesting that reduced IL-6 production in this model might be responsible for compromised macrophage mobilization in *Nelfb* KO mice. To test this possibility, *Nelfb* KO mice were administrated exogenous IL-6, which partially rescued the impaired macrophage recruitment phenotype (Fig. 7f). Collectively, these results demonstrate how by facilitating IL-6 production during inflammatory challenge, NELF serves as a positive regulator of macrophage-driven inflammation in vivo.

## Discussion
Despite overwhelming evidence for the post-initiation checkpoint in the regulation of transcription in *Drosophila* and mammalian cells, relatively little is known about the role of pause-release

mechanisms in the immune system. Here, we show that over 60% of inflammatory macrophage transcriptome is regulated primarily through Pol II proximal–promoter pausing and release, pointing at the ubiquity of post-initiation control of macrophage activation, analogous to previously described observations during certain processes such as organism development. Importantly, our data revealed unexpected modes of regulation and function of the NELF complex during macrophage activation that could not be predicted based on the existing knowledge obtained from cells of non-immune lineages. In resting macrophages, NELF is broadly associated with paused Pol II (Supplementary Fig. 7d), and its deletion did not lead to broad transcription derepression or "bursts" in the absence of activation signals resulting in CDK9 recruitment. In response to TLR signaling, the NELF complex is rapidly and globally evicted from chromatin in a stimulus-dependent yet CDK9-independent manner, releasing the "brake" on paused genes. Notably, as NELF dismissal in macrophages occurs at the global level and is not restricted to LPS-inducible genes, we envision that it likely poises genes in a nonspecific manner for subsequent activation, licensing them for subsequent signal-specific action by P-TEFb, DSIF and other regulators. Functionally, NELF promoted macrophage inflammatory gene transcription, in part, by attenuating AP-1-dependent expression of a key anti-inflammatory cytokine IL-10 (Supplementary Fig. 6) and, in part, by directly constraining the expression of NELF+ inflammation inhibitors. Thus, NELF behaves as a multifunctional regulator of macrophage transcriptome to modulate the outcomes of macrophage-mediated inflammatory responses.

Interestingly, promoter-proximal pausing was highly correlated with PU.1 occupancy around the TSS regions. PU.1 was previously implicated as a pioneer factor at macrophage-specific enhancers that helps establish and maintain open chromatin environment[29,38,39]. Indeed, PU.1 is essential for macrophage lineage specification and for the expression of a wide array of both constitutively expressed and inducible genes[23,40,41]. Previous global analysis of PU.1 distribution by ChIP-seq demonstrated that approximately 80% of DNA-associated PU.1 is enriched at distal regulatory elements with the remaining 20% occupying TSS regions[29] yet, functions of TSS-associated PU.1 remain poorly understood. We propose that PU.1 may aide in retaining paused Pol II near promoters yet detailed mechanisms underlying PU.1's action in this context await future investigation. ETS family members have been shown to function in the vicinity of core promoter elements helping to recruit basal transcriptional machinery[42], hence, it is plausible that PU.1 directly or indirectly interacts with PIC components to maintain the paused status of Pol II.

In recent years, components of basal machinery involved in Pol II pausing and early elongation have been studied extensively using in vitro biochemical approaches, in invertebrates such as fission yeast and in *Drosophila*[13]. In higher organisms including mammals, the immune system demands both rapid and exquisitely accurate transcriptional responses to internal and environmental triggers, which makes it ideal for investigating the mechanism underlying transcriptional control[43,44]. Indeed, it is well-appreciated that rapid activation of macrophage transcriptome upon infectious challenges is essential for eliciting adequate innate immunity, however, how much of it occurs during early elongation checkpoint remains unclear. What dictates stable Pol II pausing in macrophages? What are the behaviors and functions of key pausing factors during macrophage activation and how they compare to their canonical functions derived from biochemical studies? For example, in vitro, co-incubation of NELF and CDK9 abolishes NELF binding to pausing complex[32] implicating CDK9-mediated phosphorylation in NELF release. Whether this model uniformly applies to

different cellular contexts is unclear, and there has been a precedent of CDK9-independent regulation of NELF[45,46]. Our data on NELF dismissal during acute macrophage activation call into question the role of CDK9 in this process. Indeed, both pharmacological and genetic CDK9 loss-of-function experiments point to CDK9-independent LPS-induced NELF release. Nevertheless, paralysis of TLR4 signaling via MyD88 and IKKs effectively abolished inducible NELF dissociation, linking canonical TLR signaling to pause-release. Thus, our study in macrophages illustrates that the mechanisms of post-initiation transcriptional control in different cell types might be tailored to specific signaling wiring and functional needs of a given cell.

We have previously shown that post-initiation steps of the transcription cycles including pause-release and productive elongation are targeted by the well-known inhibitors of macrophage-driven inflammation such as GR[9,21,26]—a finding with clear therapeutic implications. However, the coherent picture of the regulatory principles underlying transcription elongation in macrophages has been lacking and, consequently, the specific impact of NELF on an acute inflammatory response is debated. A critical role in pausing implies that NELF represses "leaky" Pol II read-through; hence, NELF+ genes in a NELF KO would lose this checkpoint and display higher basal expression or augmented transcriptional response to an inducer. An equally plausible scenario, however, is that NELF "concentrates" transcriptionally active initiated Pol II to elicit a rapid and potent response to a stimulus; if so, loss of NELF may attenuate gene induction. Here, our comprehensive genomic and genetic analyses of NELF actions in macrophages revealed that the complex functional footprint of NELF in inflammation derives not only from its biochemical activities, but from the diversity of its targets. Indeed, NELF affects genes on each side of the inflammatory spectrum and, critically, transcription factors such as AP-1, that further propagate both pro- and anti-inflammatory networks. We report that the ultimate pro-inflammatory function of NELF is a compound of its direct inhibitory effects on NELF+ anti-inflammatory mediators as well as indirect "brakes" on the production of the AP-1 target, a broad-spectrum immuno-modulatory cytokine IL-10. These mechanistically and temporally distinct constraints on inflammation inhibitors override those that NELF imposes on paused pro-inflammatory genes. Consequently, NELF deficiency over the course of inflammatory stimulus leads to an accumulation of IL-10 that remodels the inflammatory transcriptome broadly dampening inflammatory gene expression in cultured macrophages and yielding a hypo-inflammatory phenotype in vivo. Interestingly, similarly dominant effects of IL-10 in genetic systems when both pro- and anti-inflammatory arms are affected have been documented for *Fos* deficiency[47]. Basal transcription factors and chromatin regulators are increasingly viewed as therapeutic targets in a wide range of diseases ranging from autoimmunity to cancer. Identification of NELF as a key permissive factor in inflammation needs to be considered when evaluating pause-release machinery in future drug design efforts.

## Methods

**Mice.** The laboratory animal facility has been accredited by AAALAC (Association for Assessment and Accreditation of Laboratory Animal Care International) and the IACUC (Institutional Animal Care and Use Committee) of Tsinghua University approved all animal protocols used in this study. All mice were housed in isolated ventilated cages (maxima six mice per cage) barrier facility at Tsinghua University. The mice were maintained on a 12/12-h light/dark cycle, 22–26 °C, 40–70% humidity with sterile pellet food and water ad libitum. *Nelfb*$^{fl/fl}$ mice were generated[48]. Mice with myeloid-specific deletion of *Nelfb* (*Nelfb*$^{fl/fl}$, *Lyz2Cre*) were generated by crossing *Nelfb*$^{fl/fl}$ mice to *Lyz2Cre* mice. *Myd88*$^{-/-}$ mice were purchased from the Jackson Laboratory. Age and gender matched mice were used for experiments.

**Cell culture and reagents**. Murine BMDM were obtained[49] and maintained in DMEM supplemented with 10% fetal bovine serum (FBS) and 10% L929 cell supernatant as conditioned medium providing macrophage colony stimulating factor. Cell culture grade LPS (*Escherichia coli* 0111:B4) and Pam3CSK4 were purchased from InvivoGen and used at a concentration of 10 ng/ml. SP600125 and SB203580 were purchased from Selleck, Bay 11-7082 was obtained from Sigma-Aldrich and flavopiridol was purchased from Santa Curz Biotechnology. All inhibitors were added 0.5 h prior to LPS addition and were present throughout LPS exposure.

**Chromatin immunoprecipitation (ChIP) assay and ChIP-seq**. For Pol II, NELF-E, CDK9, Fos, Jun, PU.1 ChIP assays, cells were used as indicated and left untreated or stimulated with LPS (10 ng/ml) as indicated in Figure legends. $10-25 \times 10^6$ cells per condition were fixed in 1% methanol-free formaldehyde (Thermo Scientific) for 5 min at room temperature followed by quenching with 125 mM glycine for another 5 min. Nuclear extracts were prepared[50] and chromatin DNA was sonicated to an average size of 300 bp using a Bioruptor (Diagenode). For immunoprecipitations, antibodies against Pol II (sc-9001×, Santa Cruz), NELF-E (F9 sc-377052, Santa Cruz, or 10705-1-AP, Proteintech™), CDK9 (H-169 sc-8338, Santa Cruz), PU.1(T-21 sc-352, Santa Cruz), c-Fos(9F6 2250, Cell Signaling Technology), and c-Jun(60A8 9165, Cell Signaling Technology) were used. After purification, immunoprecipitated DNA was analyzed by RT-qPCR (primer sequences are listed in Supplementary Data 1) and relative occupancies were normalized to input DNA. For Pol II and CDK9 ChIP-seq, 10 ng of DNA was ligated to adapters and 100–300 bp DNA fragments were purified to prepare libraries at Weill Cornell Epigenomics Core. For NELF-E ChIP-seq, libraries were prepared using the NEBNext Ultra II Library Prep Kit for Illumina (NEB), size selected (150–250 bp) and PCR-amplified for 15 cycles. The libraries were sequenced at the Weill Cornell Epigenomics Core using HiSeq2500.

**RNA-seq**. BMDM RNA was isolated using Qiagen RNA-easy kit. Total RNA was polyA enriched and Illumina-compatible sequencing libraries were prepared with TruSeq mRNA-Seq sample preparation kit (Illumina). Quality control of RNA and libraries was performed with the BioAnalyzer 2100. Pair-end sequencing was performed at the Weill Cornell Epigenomics Core using HiSeq2500.

**PRO-seq**. PRO-seq experiments were performed to detect de novo transcripts[51]. BMDM from WT and *Nelfb* KO mice were treated with or without LPS for 0.5 h, then BMDM nuclei ($40-50 \times 10^6$ cells per condition) were isolated by swelling cells for 5 min in ice-cold douncing buffer (10 mM Tris-HCl pH 7.4, 300 mM sucrose, 3 mM CaCl₂, 2 mM MgCl₂, 0.1% TritonX-100, 0.5 mM DTT, 1 × PIC, 4 units/ml RNase inhibitor) on ice, then gently homogenized and pelleted. The nuclei were washed twice in ice-cold douncing buffer and resuspended in storage buffer (10 mM Tris-HCl pH 8.0, 25% glycerol, 5 mM MgCl₂, 0.1 mM EDTA, 5 mM DTT) at the density of $1 \times 10^7/100\ \mu l$, and stored at $-80\ °C$. Nuclear run-on experiments were carried out with 2-biotin labeled NTP. Briefly, preheat 2× nuclear run-on buffer (10 mM Tris-HCl pH 8.0, 5 mM MgCl₂, 1 mM DTT, 300 mM KCl, 1% Sarkosyl, 2 mM biotin-11-C/UTP, 20 mM A/GTP, 0.8 units/µl RNase inhibitor) to 37 °C. Next, $10 \times 10^6$ nuclei in 100 µl of storage buffer were mixed with 100 µl of 2× nuclear run-on buffer and incubated at 37 °C for 3 min. Reaction was terminated by TRIzol addition, and RNA was isolated and fragmented by base-hydrolysis with 0.2 N NaOH on ice to an average size of 40–100 nt. Nascent RNA was isolated with magnetic streptavidin-coated beads (Invitrogen, 112.06D). The 5′ ends of RNA products were repaired by incubating with RppH (NEB, M0356S) and PNK (NEB, M0201), followed by another round of biotin-streptavidin affinity purification. RNA was prepared for constructing library with NEBNext Multiplex Small RNA Library Prep Set for Illumina (NEB, E7300). Briefly, RNA was ligated to 3′ DNA adapter by incubating for 18 h at 16 °C, hybridized with the reverse transcription primer, and ligated to 5′ RNA adapter by incubating at 25°Cfor 2 h then at 16 °C for 18 h. RNA was reverse transcribed and PCR-amplified (15 cycles). Quality control of RNA and libraries was performed with the BioAnalyzer 2100. Pair-end sequencing was performed with Hiseq X-ten.

**Immunoblotting**. Whole-cell lysates were prepared by direct lysis in sodium dodecyl sulfate (SDS) loading buffer. For immunoblotting analysis, lysates were separated by 10% SDS polyacrylamide gel electrophoresis and transferred to a polyvinylidene fluoride membrane (Millipore) for probing with specific antibodies. Antibodies against p38 (sc-535), PU.1 (sc-352) were purchased from Santa Cruz Biotechnology. Antibody against NELF-E (10705-1-AP) was from Proteintech™. Antibody against NELF-B (ab167401) was from Abcam. All the other antibodies were obtained from Cell Signaling Technology. Relative density of blotting bands was quantified using Image J (v1.52a).

**Preparation of chromatin-associated fraction**. Chromatin-associated fraction was prepared using stepwise fractionation protocol[31]. Briefly, cells were rinsed twice with ice-cold PBS, scraped into microcentrifuge tubes, and pelleted at 1500×g, 4 °C for 5 min. Cells were resuspended in 5× packed cell volume (PCV) of ice-cold buffer A (10 mM HEPES pH 7.9, 1.5 mM MgCl₂, 10 mM KCl, 1 mM DTT, 0.5 mM PMSF, 1× protease inhibitor cocktail) for 1 min and centrifuged at 250×g, 4 °C for

2 min. The cell pellets were resuspended in equal volume of Buffer A (with 0.5% NP-40) for 1 min to break down the cell membrane and nuclei pelleted at 2500×g, 4 °C for 2 min. The nuclei were extracted twice by incubating with 1× PCV of low-salt buffer (10 mM HEPES pH 7.9, 1.5 mM MgCl₂, 75 mM KCl, 0.5% NP-40, 1 mM DTT, 0.5 mM PMSF, 1× protease inhibitor cocktail) on ice for 10 min followed by centrifugation at 5000×g, 4 °C for 2 min. The low-salt extracted nuclei (containing chromatin-associated factors) were boiled with 5.6× PCV of 1× SDS-loading buffer to generate chromatin-associated fraction for immunoblotting. Antibodies for NELF-B (Abcam, ab167401), NELF-E (Proteintech, 10705-1-AP), Pol II S2P (Abcam, ab5095), NF-κB p65 (C22B4, Cell Signaling Technology, 4764) were used to detect chromatin-associated proteins. Actin levels as detected by an anti-actin antibody (ABclonal, AC026) served as loading control.

**Reverse transcription and qPCR**. RNA was extracted from whole cell lysates with a Total RNA Purification Kit (GeneMark) and reverse-transcribed to cDNA with Moloney Murine Leukemia Virus Reverse Transcriptase. qPCR was performed in triplicate with an ABI StepOnePlus thermal cycler. Threshold cycle numbers were normalized to triplicate samples amplified with primers specific for glyceraldehyde-3-phophate dehydrogenase (*Gapdh*). Primer sequences are listed in the Supplementary Data 1.

**Enzyme-linked immunosorbent assay (ELISA)**. Secreted cytokines were measured after LPS treatment by ELISA. IL-10, IL-6, TNF and IL-12 (p40) were quantified by using the Mouse IL-10 ELISA MAX™ Deluxe Set (BioLegend, 431414), Mouse IL-6 ELISA Set (BD, 555240), Mouse TNF (Mono/Mono) ELISA Set (BD, 555268), and Mouse IL-12 (p40) ELISA Set (BD, 555165) according to the manufacturers' instructions.

**LPS-induced peritonitis and IL-6 administration**. Peritonitis was induced by intraperitoneal injection of 100 ng LPS/mouse in 500 µl PBS[52]. Nine hour post-injection, mice were sacrificed by carbon dioxide exposure, and peritoneal cavities were washed with 5 ml PBS. Numbers of total peritoneal cells were counted using a hemocytometer. For IL-6 rescue experiment, in addition to LPS administration, mice were intraperitoneally injected with 40 ng IL-6/mouse in 500 µl PBS. Nine hour post-injection, mice were sacrificed, and peritoneal cells were isolated and analyzed by FACS.

**Isolation of resident peritoneal macrophages**. Resident peritoneal macrophages were prepared[53]. Briefly, peritoneal exudate cells were washed out with ice cold PBS. After washing once with PBS, peritoneal cells were resuspended in DMEM supplemented with 10% FBS and 1% P/S. Cells were then allowed to adhere for overnight in petri dishes at 37 °C. Non-adherent cells were removed by gently washing three times with warm PBS. The adherent cells were used as peritoneal macrophages.

**Flow cytometry**. After removing red blood cells with Ack lysis buffer, cells were stained with the following fluorescence-conjugated antibodies for 30 min at 4 °C: CD45R-APC/Cy7 (BioLegend, clone, dilution 1:400, Cat 103116), CD11b-PE/Cy7 (BioLegend, clone M1/70, dilution 1:400, Cat 101216), F4/80-APC (eBioscience, clone BM8, dilution 1:400, Cat 17-4801-82), Ly6G-PerCP/Cy5.5 (BD, clone 1A8, dilution 1:400, Cat 560602), Ly6C-PE (BD, clone AL-21, dilution 1:200, Cat 560592), CD11c-PE/Cy7 (eBioscience, clone N418, dilution 1:400, Cat 25-0114-82), and Siglec F-BV421(BD, clone E50-2440, dilution 1:400, Cat 562681). Cells were washed three times and analyzed on FACS Fortessa flow cytometer (BD Biosciences). Further analysis was implemented using Flowjo software (Treestar). Cell populations were defined as follows: macrophages in peritoneal cavities and spleen: CD45+CD11b+F4/80+; macrophages in bronchio-alveolar lavage fluid (BALF): CD45+CD11c+SiglecF+; monocytes: CD45+CD11b+Ly6C+; neutrophils: CD45+CD11b+Ly6G+.

**Sequence data alignment, visualization, and quantification**. Pol II ChIP-seq, CDK9 ChIP-seq, NELF-E ChIP-seq and PRO-seq data (the reverse reads from paired-end data) were collected. Adapter sequences were trimmed from the ends of reads by Cutadapt (V1.14), and the reads that failed to pass the quality control ($Q > 10$) were discarded. PU.1 ChIP-seq data set was downloaded from NCBI GEO DataSet under the GEO accessions: GSE38379 (GSM940924). SRA files were converted to fastq files using fastq-dump included in SRA toolkit. ChIP-seq and PRO-seq reads in fastq files were aligned to mouse genome (UCSC mm10) using Bowtie (version 1.1.2)[54] to generate alignment files of uniquely mapped reads with maximum allowed mismatch of 2 (-m 1 -n 2) for each ChIP-seq data set, and PRO-seq reads aligned with short seed length (-l 10). ChIP-seq reads aligned to genome were extended to 150 bp from their 3′ end for further analysis. ChIP-seq and PRO-seq alignment files were visualized as bedgraph files with normalized reads count (per 10 million reads) at 1 bp resolution, which were generated by using HOMER (v4.7.2)[38]. Bedgraph files were loaded to IGV (Integrative Genomics Viewer, v2.3), and individual gene tracks were obtained as snapshots from IGV.

RNA-seq data were collected and pair-end reads were aligned to mouse genome mm10 using TopHat 2.1.0[55] with the parameters -i 70 -g 1–no-novel-indels–coverage-search, and only uniquely mapped reads were preserved.

As PRO-seq reads are complementary to in vitro nuclear run-on products, the beginning of PRO-seq reads reflects the actual transcription-active site. Therefore, we shortened PRO-seq reads to 1 bp for quantification, and used the reads aligned to anti-sense strand as the transcription-active signals in sense strand for each gene. To determine ChIP-seq and PRO-seq signals around TSS, we first counted ChIP-seq extended reads (-fragLength 150) and PRO-seq shortened reads (-fragLength 1 -strand -) every 10 bp from TSS to both upstream 1 kb and downstream 2 kb regions for each gene by using annotatePeaks.pl program in HOMER. The output counting matrixes were next used to generate signal heat map around TSS regions by Cluster Treeview 1.1.6. The average signals around TSS regions were calculated as the average reads count per bin (10 bp) per gene.

Upregulated and downregulated genes in PRO-seq were defined as normalized PRO-seq reads (RPKM + 1) fold changes (Nelfb KO/WT) ≥ 1.4 for up-regulated genes and fold changes (Nelfb KO/WT) ≤ 0.6 for down-regulated genes. Counting PRO-seq reads for each gene was implemented using annotatePeaks.pl program in HOMER.

Peaks of PU.1 ChIP-seq and NELF-E ChIP-seq were called by findPeaks program in HOMER (FDR < 0.001).

**RNA-seq data analysis**. For coverage of mapped RNA-seq reads in transcripts, the expression level of each gene transcript was calculated as the average fragments count of three biological replicates, which was subsequently normalized as the fragments per kilobase of transcript per million mapped reads (FPKM). BMDM-expressed genes were defined as genes with FPKM ≥ 1 in resting BMDM. Both expression level (FPKM) and differential gene expression between experimental conditions was identified by using Cuffdiff program in Cufflinks 2.2.1[56]. Genes with q-value < 0.05 and (FPKM + 1) fold changes ≥ 1.5 between WT resting and LPS treated (1 h) BMDM were regarded as LPS-inducible (n = 449). Upregulated and downregulated genes in Nelfb KO BMDM were identified as p value < 0.05, (FPKM + 1) fold changes (Nelfb KO/WT) ≥ 1.4 for upregulated genes and (FPKM + 1) fold changes (Nelfb KO/WT) ≤ 0.6 for downregulated genes. LPS-induced (n = 83) and LPS-suppressed (n = 161) genes at 0.5 h were identified with p value < 0.05 and (FPKM + 1) fold change (LPS 0.5 h/0 h) ≥ 1.4 for induced genes and fold change (LPS 0.5 h/0 h) ≤ 0.6 for suppressed genes. "Superinduced" and "more-suppressed" genes in Nelfb KO BMDM were defined as LPS-induced or LPS-suppressed (0.5 h) genes in the WT, which were further upregulated or downregulated in Nelfb KO BMDM.

**PI calculation and gene categorization**. Refseq gene annotation for mm10 was obtained from UCSC table browser, and genes from mitochondrial and random chromosomes were filtered out. 24030 total unique genes were defined as the total longest variant for each gene with unique gene symbol. To precisely calculate Pol II PI for genes with sufficient gene body length, we chose unique genes longer than 1 kb for further Pol II distribution pattern analysis in Pol II ChIP-seq replicate 1.

We defined TSS region for each gene as −250 bp to +250 bp relative to TSS, and the gene body was defined as the +250 bp from TSS to TTS (transcription termination site defined by the UCSC annotation, which means cleavage and poly-adenylation site). Using the findPeaks program in HOMER, we set 500 bp window to search Pol II enriched region in Pol II ChIP-seq replicate 1 data in resting BMDM, and obtained the threshold value of reads count for statistically Pol II enriched TSS regions (FDR < 0.001) as 38. Therefore, in 10076 BMDM-expressed genes (≥1 kb), 8083 genes with more than 38 reads in TSS regions are defined as genes with Pol II+ TSS regions, and the other 1993 genes are defined as genes with Pol II− TSS regions.

For 8083 genes with Pol II+ TSS regions, Pol II PI were calculated as the ratio of Pol II ChIP-seq reads density (reads per 1 kb) in TSS regions to gene body regions in resting and LPS activated BMDM. Genes with PI ≥ 3 were considered as highly paused and genes with 1.5 ≤ PI < 3 were defined as moderately paused. Genes with PI < 1.5 were defined as non-paused along with genes with Pol II− TSS region.

Similar to gene groups categorized by Pol II ChIP-seq defined PI, we used PRO-seq dataset to re-define pausing status in resting BMDM. PRO-seq defined PI were calculated as the ratio of shortened (1 bp) PRO-seq reads density (reads per 1 kb) in TSS regions to that in gene body regions. Using the same method and criteria as Pol II ChIP-seq defined pausing status, among 10076 BMDM-expressed genes (≥1 kb), 8809 genes with more than 13 shortened PRO-seq reads in TSS regions are defined as genes with significantly (FDR < 0.001) transcription-active TSS regions, whereas the other 1267 were defined as genes with transcription-inactive TSS regions. Among 8809 genes with transcription-active TSS regions, 7562 genes with PRO-seq PI ≥ 3 were considered as highly paused and 777 genes with 1.5 ≤ PRO-seq PI < 3 were defined as moderately paused. 470 genes with PRO-seq PI < 1.5 along with 1267 genes with transcription-inactive TSS regions were defined as non-paused.

**Assessing NELF occupancy at TSS regions**. We used two methods to classify NELF occupancy at TSS regions. (1) We used DiffBind (v2.12.0)[57] to assess differential NELF occupancy at TSS regions. BED files for TSS regions of all genes were used as peak region input, and sorted BAM files for two replicates of NELF ChIP-seq were used as input for reads counting. We used EdgeR mode in DiffBind to identify differential binding, and considered fold change (LPS 0.5 h/0 h) of NELF−E ChIP-seq reads count less than 0.5 and p value < 0.05 as threshold for significant NELF dismissal in TSS regions. (2) Peak calling in ChIP-seq is often utilized to identify differential occupancy. We used findPeaks program in HOMER to call NELF-E ChIP-seq peaks, with LPS 0.5 h condition as background and LPS 0 h condition as enrichment, to identify genes with TSS regions showing significant NELF dismissal upon LPS stimulation.

**Motif enrichment analysis**. To identify enriched known transcription regulatory elements in target TSS regions, we used findMotifsGenome.pl program in HOMER to find enriched motifs. Sequences from TSS regions of these genes were used as inputs and randomly extracted sequences from the genome were used as background.

**Statistics**. Statistical analysis was performed by using two-tailed Student's t test or other model where indicated. p < 0.05 was taken as statistically significant unless otherwise indicated. Statistical analyses were performed using GraphPad Prism 7 and R 3.3.0.

**Reporting summary**. Further information on research design is available in the Nature Research Reporting Summary linked to this article.

## Data availability
All genomic data described herein are deposited in Gene Expression Omnibus under accession numbers GSE122292, GSE123557, GSE122300, GSE103795, and GSE123370. The source data underlying Figs. 2i, j, 4c–g, 5a, 6a, b, 6e–g, and 7a–f and Supplementary Figs. 2b–d, 4a, 4h–j, 5f, 6a, b, 6d–f, and 7a, b are provided as a Source Data file. Other data that support the findings of this study are available from the corresponding author upon reasonable request.

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

## Acknowledgements

This research was supported by Ministry of Science and Technology of China National Key Research Project 2015CB943201 (X.H.), National Natural Science Foundation of China (31725010 and 31821003 to X.H.), funds from Tsinghua-Peking Center for Life Sciences (X.H.), and funds from Institute for Immunology at Tsinghua University (X.H.). M.A.S. was supported by the NIH NIDDK Diversity Supplement 3R01DK099087-01A1S1 and the MSTP grant T32GM007739 from the NIH NIGMS to the Weill Cornell/Rockefeller/Sloan-Kettering MD-PhD Program. D.A.R. was supported by the NIH NIAMS T32AR007281. I.R. was supported by the grants from NIH R01DK099087 and R01AI148129, the Rheumatology Research Foundation Research Grant, the DOD CDMRP PR130049 Research Award and The Hospital for Special Surgery David Rosensweig Genomics Center.

## Author contributions

L.Y. and B.Z. designed the research, performed the experiments, analyzed the data, and wrote the paper. M.A.S. performed NELF-E ChIP-seq and RNA-seq experiments. D.D. performed NELF-E ChIP-seq and Bay 11-7082 ChIP-qPCR experiments. Y.S. and M.C. performed Pol II ChIP-seq experiments. D.A.R. and B.T. performed CDK9 ChIP-seq experiments. Z.G. generated CDK9-KD iBMDM. X.Z. generated PU.1-*Tnf*-mut iBMDM. R.L. generated *Nelfb*<sup>fl/fl</sup> mice. Y.C. performed computational genome-wide data analysis and processing. I.R. and X.H. conceptualized the project, designed the research, supervised the experiments, and wrote the paper.

## Competing interests

The authors declare no competing interests.
