## [Peer Review File · Nature Communications]

Reviewers' comments:

Reviewer #2 (Remarks to the Author):

In this article by Yu et al., the authors investigate the role of promoter proximal pausing in immune activation in macrophages. They first profile NELF and Pol II to establish that many genes are paused in macrophages and, as expected, paused genes are associated with NELF occupancy. Genes with the highest levels of pausing appear to be enriched for ETS-family binding sites within their promoters, which they attribute to PU.1 by analyzing ChIP-seq data or performing targeted functional assays at the *Tnf* gene. They then examine changes in pausing and NELF occupancy after inducing for 30 minutes with LPS. They find the pausing indices decrease somewhat at LPS induced genes which is coincident with a decrease in NELF signal at these promoters. Interestingly, they find that NELF is transiently reduced at all promoters after LPS treatment and that this appears to be independent of P-TEFb. Knockout of *NelfE* gene showed little change in resting cells, but showed that many of the LPS inducible genes were hyperactivated. They chose to focus on *IL10*, an a presumed indirect target of NELF, and show that the AP1 complex that regulates it and whose components (*Fos* and *Jun*) are direct NELF target are also hyperactivated. Since *IL10* and several of the other hyperactivated genes are inflammation inhibitors, they suggest that NELF positively affect immune response and show macrophage stimulation data in the *Nelf* KO to support this. There are some interesting points made in this work – of note is the P-TEFb-independent removal of NELF from chromatin, and the potentially interesting interruption of this gene signaling network through transcription regulation. However, there are also several weaknesses outline below that need to be addressed.

Comments:

1. The statement on lines 82-86 seems to contradict itself. They mention that post-initiation regulation of transcription is underappreciated in immune function, but then cite at least 6 papers that have been addressing this. They should be more explicit about what has and has not been shown in regarding this area.
2. The title of the first results section, "The majority of macrophage transcriptome is regulated by Pol II promoter- proximal pausing" is a bit of an overstatement. Finding that 70% of genes have evidence of promoter proximal pausing is not the same as showing that their levels are effectively regulated through pausing. For instance, gene that have paused Pol II can still be regulated at the stage of initiation which results in equivalent change in both the promoter and gene regions.
3. They report 76% of active genes as being paused (by ChIP-seq). Are the numbers comparable from the PRO-seq data?
4. How many genes are regulated in response to LPS? Only percentages are used until Figure 5 where 83 genes appear to be induced. This number seems somewhat low.
5. What about down regulated genes? Are there appreciable number of down regulated genes and does their NELF occupancy change? How are they affected in the *Nelf* KO?
6. The description of the results on lines 167-172 seem to be a bit of an overstatement. For instance they describe "some" additional loading of Pol II at the promoter and a "massive" increase in genes. However, the data does not really support this. There appears to be an increase in both promoter and the gene with fold change in genes only slightly higher than promoters, which result in the modest changes in PI shown in figure 3c.
7. Changes in pausing indices after LPS should be quantified and reported for PROseq data. ChIPseq data, as analyzed may not represent only polymerases moving into the gene since they will also include signal from the upstream antisense polymerase. This is critical to support that the change in transcription of these genes is primarily post-initiation. At the current depth of analysis,

I am not convinced that the regulation is entirely or even mostly post-initiation.

8. PRO-seq composite profile in figure 3D looks odd. Perhaps it is overly smoothed? The signal should not start at -250 relative to the TSS, and the peak shapes look like no other I have seen with this type of directional genomic nascent RNA data.

9. Since the NELF reduction on the chromatin is transient, and loss of NELF leads to hyperactivation of some genes, does this suggest that transient NELF inactivation could allow for bursts of transcription activity? This is not discussed.

Minor Comments:

1. Group 3 (non-paused genes) should be defined in the main text, not just the figure legend.

Reviewer #3 (Remarks to the Author):

Re. The majority of macrophage transcriptome is regulated by Pol II promoter-proximal pausing

The authors used the "total longest variant" to define each gene, which is not justified in the methods. In order to accurately define a pause index, the transcription initiation/start site must be accurate.

Although pausing is wide-spread and found at $\frac{3}{4}$ of the BMDM transcriptome, this is not appreciably different than the fraction of active and paused genes in most cell types. The same is true of NELF occupancy. The correlation between pausing and NELF is also well-characterized.

The authors fail to mention the fraction of paused genes that contain NELF, so the significance of the distribution of NELF at LPS-activated genes (Figure 3g) is impossible to determine.

The authors do not separate their PRO-seq data by strand, which makes their heatmaps and metagene profiles misleading.

Re. PU.1 facilitates promoter-proximal pausing in macrophages

The motif analysis and ChIP-seq analysis is descriptive and shows that PU.1 is associated with the expression of many genes, which is expected based on its expression and role in lineage determination.

Depletion of PU.1 and ChIP-qPCR at two genes is a good first step in establishing the role of PU.1 in transcription, but it falls short. I am unclear what the authors mean by "facilitates promoter-proximal pausing". I interpret this to mean that PU.1 directly regulates pausing in some way. However, if the role of PU.1 is to recruit Pol II or facilitate initiation, then the ChIP-qPCR results would be identical. Therefore, this experiment to determine the functional role of PU.1 is not conclusive and doesn't support the section heading.

Re. NELF dismissal associated with Pol II pause-release is a key feature of macrophage activation

It is unclear how the authors use statistics to quantitate LPS activated genes. Figure 3a and 3b are heat maps and metagene profiles. However, Figure 3c,d, and e support the author's conclusion that LPS-mediated activation facilitates efficient pause release or transcription through the pause site. Assuming that LPS-mediated activation is through NF- κ B, this phenomenon has been previously demonstrated (PMID: 23523369), but this work is not acknowledged.

The authors do not perform the appropriate controls and statistics to conclude that NELF dissociates from promoter proximal regions. It would be nice to see some routine statistics like DiffBind and qPCR with % inputs for a subset of NELF binding sites.

Re. NELF dissociation from chromatin of activated macrophages is global yet transient

This section is very interesting, but under-developed. I interpret this section to conclude that there is no specificity to NELF-depletion, but the authors need to quantitate changes in NELF occupancy globally with a differential binding statistics package. Is the degree to NELF depletion constant at all regions in the genome or are some peaks more susceptible? What did the authors do rule out any specificity to conclude a non-specific global decrease.

Re. NELF controls an AP-1 dependent transcriptional circuit to target IL-10

Figure 6a, it is unclear if NELF depletion affects the 0hr LPS time point Il10 levels or whether this is normalized out.

I agree with the author's conclusions that NELF and LPS negatively regulates AP1 subunits. However, the conclusion that AP1 is directly regulating key immune regulators downstream is based upon searching for AP1 motifs, so this conclusion is not supported by rigorous experimental or analytical evidence.

In general the software used for analyses is not cited and the analyses are not reproducible based on the methods descriptions.

Nature Communications Point-to-point response:

“Negative elongation factor complex enables macrophage inflammatory responses by controlling anti-inflammatory gene expression” by Li Yu, Bin Zhang, Dinesh Deochand, Maria A. Sacta, Maddalena Coppo, Yingli Shang, Ziyi Guo, Xiaomin Zeng, David A. Rollins, Bowranigan Tharmalingam, Rong Li, Yurii Chinenov, Inez Rogatsky, and Xiaoyu Hu

We thank the reviewers for their time and insightful and constructive comments. We are pleased that reviewers found “*interesting points made in this work*”. We have experimentally addressed the points raised by the reviewers and have substantially revised the manuscript by adding 19 panels of new figures, modifying 5 panels of current figures as well as clarifying writing. The reviewers’ points are specifically addressed below. Changes in the manuscript have been underlined.

Reviewers' comments:

Reviewer #2:

1. The statement on lines 82-86 seems to contradict itself. They mention that post-initiation regulation of transcription is underappreciated in immune function, but then cite at least 6 papers that have been addressing this. They should be more explicit about what has and has not been shown in regarding this area.

We removed the former reference #24 (Chan CH et al. BET bromodomain inhibition suppresses transcriptional responses to cytokine-Jak-STAT signaling in a gene-specific manner in human monocytes. *Eur J Immunol* 45, 287-297) as it was misplaced. Out of remaining 5 studies that implicated transcription elongation as a target of immune regulation, 3 (#9, #21, #22) were from the Hu and the Rogatsky research teams. Therefore, we are fully aware that these previous research efforts were limited to certain target genes of interest (e.g., *Cxcl1* for #21 and glucocorticoid-repressed genes for #22), whereas the general characteristics and a global role of post-initiation transcriptional control for macrophage activation remain obscure. We have edited ‘Introduction’ to flesh out this point (page 4).

2. The title of the first results section, “The majority of macrophage transcriptome is regulated by Pol II promoter- proximal pausing” is a bit of an overstatement. Finding that 70% of genes have evidence of promoter proximal pausing is not the same as showing that their levels are effectively regulated through pausing. For instance, genes that have paused Pol II can still be regulated at the stage of initiation which results in equivalent change in both the promoter and gene regions.

In this study, 76% of the BMDM transcriptome showed promoter-proximal pausing based on our Pol II ChIP-seq data sets. Importantly, PRO-seq further supported the notion that paused Pol II was transcriptionally active (Fig. 1d-e), suggesting that the majority of macrophage transcriptome presents promoter-proximal pausing features

(described on page 5). We agree that paused genes can *additionally* be regulated at the stage of Pol II initiation. Indeed, upon macrophage activation, group 1 genes displayed increased Pol II signals both in TSS regions and gene body (Fig. 3a-b; stated on page 7), which indicates that paused genes can be regulated both at the stage of initiation and pause-release.

3. They report 76% of active genes as being paused (by ChIP-seq). Are the numbers comparable from the PRO-seq data?

We thank the reviewer for raising this interesting point, which prompted us to re-analyze our PRO-seq data. The analysis showed that 83% of BMDM-expressed genes are paused according to PRO-seq signals (**new supplementary Fig. 1d**, described on pages 5-6). In addition, of the 7655 paused genes defined by Pol II ChIP-seq and 8339 paused genes defined by PRO-seq, a vast majority (7012 genes) overlap (**new Supplementary Fig. 1e**, described on pages 5-6), indicating the robustness of these assays. In summary, these two distinct methodologies independently captured near identical global promoter-proximal pausing landscape in primary macrophages.

4. How many genes are regulated in response to LPS? Only percentages are used until Figure 5 where 83 genes appear to be induced. This number seems somewhat low.

LPS-inducible genes were defined based on our RNA-seq data sets, in which the genes with $p\text{-value} < 0.05$ and (FPKM+1) fold changes (LPS 0.5 h/0 h) ≥ 1.4 were regarded as LPS-induced ($n = 83$). This number seems somewhat low, because it captures acute rapid stimulation (LPS for 0.5 h); a significantly greater number of 449 genes are upregulated upon 1 h LPS stimulation (see page 23).

5. What about down regulated genes? Are there appreciable number of down regulated genes and does their NELF occupancy change? How are they affected in the Nelf KO?

To address these questions, we re-analyzed our RNA-seq and NELF-E ChIP-seq data sets for regulatory patterns of genes downregulated by LPS treatment ($n=161$ fulfilled by criteria $p < 0.05$, LPS 0.5 h/ 0 h ≤ 0.6). A large percentage of LPS-suppressed genes displayed features of pausing as defined by high pausing index (group 1 + group 2 = 80%) and LPS induced NELF dissociation from these genes (**new Supplementary Fig. 5d**, described on page 11). However, NELF deficiency did not profoundly alter expression levels of LPS-suppressed genes (**new Supplementary Fig. 5e**, described on page 11).

6. The description of the results on lines 167-172 seem to be a bit of an overstatement. For instance they describe “some” additional loading of Pol II at the promoter and a “massive” increase in genes. However, the data does not really support this. There appears to be an increase in both promoter and the gene with fold change in genes only slightly higher than promoters, which result in the modest changes in PI shown in figure

3c.

We have edited the description as follows “displayed a dramatic increase in both Pol II loading and elongation of paused Pol II into gene bodies indicative of pause-release process” (page 7). The relatively modest changes in pausing indices defined by Pol II ChIP-seq upon macrophage activation indeed result from an increase in Pol II in both promoter and the gene body (Fig. 3b, c). Interestingly, PRO-seq data reveals more significantly decreased PI of group 1 genes in response to LPS, further corroborating pause-release (**new Fig. 3f**, described on page 8).

7. Changes in pausing indices after LPS should be quantified and reported for PROseq data. ChIPseq data, as analyzed may not represent only polymerases moving into the gene since they will also include signal from the upstream antisense polymerase. This is critical to support that the change in transcription of these genes is primarily post-initiation. At the current depth of analysis, I am not convinced that the regulation is entirely or even mostly post-initiation.

According to pausing indices derived from PRO-seq data, LPS-induced genes (n=449) were categorized into three groups, with group 1 being the highly paused under the resting state (**new Supplementary Fig. 3d**). Upon LPS stimulation, pausing indices were significantly decreased (**new Fig. 3f**, $p < 2.2e-16$, described on page 8), consistent with an augmented ratio of transcription events in gene body regions.

8. PRO-seq composite profile in figure 3D looks odd. Perhaps it is overly smoothed? The signal should not start at -250 relative to the TSS, and the peak shapes look like no other I have seen with this type of directional genomic nascent RNA data.

In the previous version, the PRO-seq signals were indeed overly smoothed by default HOMER setting, in which PRO-seq reads were treated as ChIP-seq reads by lengthening them to ~150 bp. We re-analyzed all PRO-seq data and shortened the mapped PRO-seq reads to 1 bp, which reflected actual transcription-active site (see revised **Fig. 1d, e, Fig. 3d, e**).

9. Since the NELF reduction on the chromatin is transient, and loss of NELF leads to hyperactivation of some genes, does this suggest that transient NELF inactivation could allow for bursts of transcription activity? This is not discussed.

Currently, our data sets did not detect such transcriptional ‘bursts’ upon NELF deletion. Firstly, we observe minimal global transcription disturbance in resting NELF KO BMDM. Secondly, despite of the fact that LPS induced broad NELF dissociation from promoter-proximal regions, LPS only activated a few hundred genes and a smaller subset was ‘super-induced’ in the KO compared to WT (now discussed on page 14). Therefore, we propose that NELF functions as ‘brake’ during pausing-to-elongation transition whereas productive transcription requires mobilization of some more

specifically-acting positive regulator, such as CDK9 (Supplementary Fig. 7c).

Minor Comments:

1. Group 3 (non-paused genes) should be defined in the main text, not just the figure legend.

Group 3 genes are now defined in the main text (page 5).

Reviewer #3:

Re. The majority of macrophage transcriptome is regulated by Pol II promoter-proximal pausing

The authors used the “total longest variant” to define each gene, which is not justified in the methods. In order to accurately define a pause index, the transcription initiation/start site must be accurate.

An alternative commonly used method is to include every possible TSS, as annotated by different transcript variants, for each gene. This, however, will result in more than one pausing index for a subset of genes, and it would be difficult to reconcile multiple pausing indexes with RNA-seq data, as RNA-seq analysis includes all transcripts for each gene. Considering that only a few genes have more than one well-supported TSS, we used the longest variant to assign one TSS per gene. It might be an oversimplification, but it did enable us to integrate our ChIP-seq and PRO-seq data with RNA-seq data and draw conclusions from pausing indices at a genome-wide scale.

Although pausing is wide-spread and found at $\frac{3}{4}$ of the BMDM transcriptome, this is not appreciably different than the fraction of active and paused genes in most cell types. The same is true of NELF occupancy. The correlation between pausing and NELF is also well-characterized.

We agree that the fraction of paused among active genes in most resting cell types might be similar. However, relatively little is known about the regulation of pausing landscape during immune cell activation, which was the aim of our study: 5 out of 7 main figures focus on activated macrophages.

The authors fail to mention the fraction of paused genes that contain NELF, so the significance of the distribution of NELF at LPS-activated genes (Figure 3g) is impossible to determine.

To address this concern, we analyzed the percentage of NELF⁺ genes in groups 1, 2, 3 among all BMDM-expressed genes. 32% of highly paused genes and 21% of moderately paused genes were NELF⁺ whereas only 8% of non-paused genes were NELF⁺ (**new Supplementary Fig. 1g**, see page 6). Furthermore, a considerably higher

percentage of NELF⁺ genes among paused LPS-inducible genes (50% of highly paused and 32% of moderately paused genes; Fig. 3h) compared to that in all BMDM expressed genes suggests that NELF may preferentially participate in LPS-activated transcription.

The authors do not separate their PRO-seq data by strand, which makes their heatmaps and metagene profiles misleading.

In the previous version of manuscript, PRO-seq signals were inadvertently processed by default HOMER setting, which treated PRO-seq reads as ChIP-seq reads lengthening them to 150 bp. During the revision, we re-analyzed PRO-seq data by shortening the mapped PRO-seq reads to 1 bp, which reflect the actual transcription-active sites in a complementary strand. We re-drew **Fig. 1d,e** and **Fig. 3d,e** using sense strand information (described on page 22) and modified figure legends accordingly.

Re. PU.1 facilitates promoter-proximal pausing in macrophages

The motif analysis and ChIP-seq analysis is descriptive and shows that PU.1 is associated with the expression of many genes, which is expected based on its expression and role in lineage determination.

Published reports have shown that approximately 80% of genome-wide PU.1 peaks are associated with intragenic and extragenic regions (for example, Ghisletti S *et al*, Immunity 2010, cited in the manuscript). In macrophages, PU.1 acts as an essential lineage-determining transcription factor that contributes to the selection of a large fraction of the cell-specific enhancer-like elements. PU.1 binding at macrophage-specific constitutive/poised enhancers is required for maintaining H3K4me1 and remodeling nucleosomes (Barozzi I *et al*, Mol Cell 2014 and Heinz S, *et al*, Mol Cell 2010, cited in the manuscript). Upon macrophage activation, PU.1 cooperates with stimulus-activated transcription factors to form latent enhancers, which persist and mediate a faster and stronger response upon re-stimulation (Ostuni R *et al*, Cell 2013, cited in the manuscript). When reanalyzing the published PU.1 data sets, we found that the remaining 20% of DNA-bound PU.1 was located nearby TSS. Our study describes an altogether different function of PU.1 near promoters during early elongation in maintaining the Pol II pause (Fig. 2). In the revised manuscript, we show that mutating endogenous PU.1 binding motif near the *Tnf* promoter by the CRISPR-Cas9 technique in macrophages attenuated the occupancy of Pol II and NELF (**new Supplementary Fig. 2a, b, c, d**, described on page 7), further supporting a novel role of TSS-centric PU.1 molecules.

Depletion of PU.1 and ChIP-qPCR at two genes is a good first step in establishing the role of PU.1 in transcription, but it falls short. I am unclear what the authors mean by “facilitates promoter-proximal pausing”. I interpret this to mean that PU.1 directly regulates pausing in some way. However, if the role of PU.1 is to recruit Pol II or

facilitate initiation, then the ChIP-qPCR results would be identical. Therefore, this experiment to determine the functional role of PU.1 is not conclusive and doesn't support the section heading.

In the previous version of the manuscript, what we meant by “facilitates promoter-proximal pausing” was that promoter-associated PU.1 contributed to the establishment or maintenance of promoter-proximal pausing complex by promoting Pol II and NELF occupancy, yet we didn't exclude a potential role of PU.1 in initiation. We now modified the section title to soften the claim (page 6). In addition, new data also show that disruption of the PU.1 binding core motif on the *Tnf* promoter impaired the promoter-proximal pausing (**new Supplementary Fig. 2a, b, c, d**, described on page 7, also see the point above).

Re. NELF dismissal associated with Pol II pause-release is a key feature of macrophage activation

It is unclear how the authors use statistics to quantitate LPS activated genes. Figure 3a and 3b are heat maps and metagene profiles. However, Figure 3c,d, and e support the author's conclusion that LPS-mediated activation facilitates efficient pause release or transcription through the pause site. Assuming that LPS-mediated activation is through NF- κ B, this phenomenon has been previously demonstrated (PMID: 23523369), but this work is not acknowledged.

For the first point, we defined LPS inducible genes as genes with q-value < 0.05 and (FPKM+1) fold changes of LPS 1 h/0 h \geq 1.5 (now described on page 23).

For the second point, Danko *et al* (PMID: 23523369; Mol Cell 2013) showed that TNF- α -NF- κ B pathway stimulates gene expression by reducing Pol II residence time at pause site in breast cancer cells. In our system, identifying the contribution of specific transcription factors to pause-release at individual genes was not the goal of the study. We envision that indeed, NF- κ B plays an important role in LPS-induced NELF dismissal for a subset of genes; this function is likely not limited to NF- κ B and certainly not every paused LPS-responsive gene is an NF- κ B target, given that pausing is a general phenomenon in inducible gene expression. We now include the above discussed paper in bibliography.

The authors do not perform the appropriate controls and statistics to conclude that NELF dissociates from promoter proximal regions. It would be nice to see some routine statistics like DiffBind and qPCR with % inputs for a subset of NELF binding sites.

We now added NELF-E ChIP-qPCR for *Tnf*, *Jun*, *Cited2* and *Fos* TSS regions from multiple independent experiments and statistical analysis showed significant NELF-E dismissal in the 4 genes tested (**new Supplementary Fig. 4a**, described on page 9). For NELF-E ChIP-seq data, we used DiffBind and peak calling to quantify NELF dismissal.

LPS stimulation resulted in NELF dissociation from approximately half of NELF⁺ genes (55% by Call peak and 46% by DiffBind) (**new Supplementary Fig. 4c**, described on page 9).

Re. NELF dissociation from chromatin of activated macrophages is global yet transient

This section is very interesting, but under-developed. I interpret this section to conclude that there is no specificity to NELF-depletion, but the authors need to quantitate changes in NELF occupancy globally with a differential binding statistics package. Is the degree to NELF depletion constant at all regions in the genome or are some peaks more susceptible? What did the authors do rule out any specificity to conclude a non-specific global decrease.

We quantified NELF dismissal using DiffBind and peak calling methods. Significant NELF dismissal was defined as fold change LPS 0.5 h/0 h of NELF-E ChIP-seq read density in TSS regions < 0.5 and p-value < 0.01 for DiffBind, and TSS regions with peaks called by using NELF-E ChIP-seq at LPS 0 h as sample and LPS 0.5 h as background. We found that about half of TSS regions of NELF⁺ genes showed significant NELF dismissal (**new Supplementary Fig. 4c**, described on page 9). We also quantified fold changes (LPS 0.5 h/0 h) of NELF-E ChIP-seq reads density in TSS regions and showed in **new Supplementary Fig. 4d** that the majority of NELF⁺ genes displayed loss of NELF occupancy upon LPS stimulation (**new Supplementary Fig. 4d, e**, described on page 9). The above analyses supported the notion that NELF dismissal is global and not restricted to LPS-induced transcriptome, although we do not rule out the possibility that some sites are more susceptible than others as the reviewer suggested.

Re. NELF controls an AP-1 dependent transcriptional circuit to target IL-10

Figure 6a, it is unclear if NELF depletion affects the 0hr LPS time point *Ii10* levels or whether this is normalized out.

We used $2^{-\Delta Ct}$ to calculate the mRNA expression level relative to *Gapdh*. The *Ii10* expression is largely stimulus (LPS)-dependent, which makes it difficult to assess the differences at the baseline level. Re-assessment of basal *Ii10* expression data show that NELF depletion modestly yet consistently upregulated basal *Ii10* expression, which may be driven by increased levels of *Fos* and *Jun* (**new Supplementary Fig. 6a, b**, described on pages 11-12).

I agree with the author's conclusions that NELF and LPS negatively regulates AP1 subunits. However, the conclusion that AP1 is directly regulating key immune regulators downstream is based upon searching for AP1 motifs, so this conclusion is not supported by rigorous experimental or analytical evidence.

To experimentally address this question, we performed Fos and Jun ChIP-qPCR to demonstrate that LPS stimulation markedly enhanced AP-1 occupancy at the predicted AP-1 binding motifs associated with the genes encoding key immune regulators, e.g., *Ili10*, *Ili1b*, *Tnfaip3*, thus confirming that these genes are indeed AP1 targets in macrophages (**new Supplementary Fig. 6d, e**, described on page 12).

In general the software used for analyses is not cited and the analyses are not reproducible based on the methods descriptions.

To enhance clarity, we added more detail to methods, especially related to data analysis. For example, PRO-seq analysis is now described in detail on page 22, LPS-induced and LPS-suppressed genes are now better defined on page 23, pausing indices defined with PRO-seq data sets are depicted on page 24, and identification of NELF-occupied TSS regions using DiffBind and peak calling is described on page 24-25. We also clearly cited the literature regarding the software used in the analysis such as newly added references #53 for Bowtie, #54 for TopHat, #55 for Cufflinks and #56 for DiffBind.

Reviewers' comments:

Reviewer #2 (Remarks to the Author):

Remarks to the Author Much of the data appears to have been reanalyzed and the results are improved. The observations of regulation through pause release, transient and P-TEFb independent dismissal of NELF, and NELF-dependent attenuation of the transcription factors that activate Il10, are more solidified and are significant findings for the gene regulation field.

Major comments remaining are below:

As mentioned by another reviewer, Pausing is widespread in most cells types examined and highly correlated with NELF occupancy. Thus the statement in the abstract, "Here, we report widespread promoter-proximal RNA polymerase II pausing in resting macrophages, marked by broad co-localization of the negative elongation factor (NELF) complex. . ." could be said about virtually any cell type and is not specific to macrophages. They really should back off from this as a significant claim in the abstract, results , and discussion.

It's pretty clear from the heatmap in figure 1d, that group 2 and 3 might be called partially as a result of the annotation list used. As pointed out by reviewer #3, using the longest annotation is not ideal. In most cases, you will end up using the most 5'-isoform of the gene. Since the TSS is then defined as +/-250 bp from the TSS, genes with more active isoforms downstream of this window will be miscategorized. It can be clearly seen in the heatmap as the apparent peak right downstream of the TSS window. Thus, many of the group 3 genes should be in group 1 or 2 if the correct TSS was used. In the end, it might not affect the results and conclusions here, but it does look like a sloppy analysis oversight in the figure.

Methods on the chromatin fractionation should for detecting NELF association should be included.

The conclusion that PU.1 "facilitates Pol II pausing" is still misleading (pointed out by reviewer #3 earlier). PU.1 regulation of genes could actually be through recruitment and initiation of the transcription machinery, making a role for pausing indirect. As stated, the sentence implies that PU.1 is involved in pausing. This cannot be concluded from the current data. This section is actually a distraction from the rest of the work and the authors should consider removing it.

Have the authors generated PRO-seq in the 0.5 hours after LPS treatment? Since less NELF is associated to TSS, it would be interesting to see if you get an expected (at least in this case) increase pause release above what is seen at 1 hour.

Reviewer #3 (Remarks to the Author):

A previous point that I do not believe was adequately addressed was the fact that 3/4 of the genome BMDM transcriptome is paused. I don't believe that this is different than any other cell type, so I am unsure why this is highlighted as being substantial or meaningful.

I am not in favor of the authors using and reporting p-values that are not multiple testing corrected in their analysis. While they do multiple testing correct in the RNA-seq by thresholding on q-value, they do not for other tests. I was unable to assess the appropriate nature of the statistics in the previous version because an adequate methods section was not provided.

The authors use TTS (transcription termination site), when they intend to use "cleavage and poly-Adenylation site"

Nature Communications point-by-point response:

“Negative elongation factor complex enables macrophage inflammatory responses by controlling anti-inflammatory gene expression” by Li Yu, Bin Zhang, Dinesh Deochand, Maria A. Sacta, Maddalena Coppo, Yingli Shang, Ziyi Guo, Xiaomin Zeng, David A. Rollins, Bowranigan Tharmalingam, Rong Li, Yurii Chinenov, Inez Rogatsky, and Xiaoyu Hu

We thank the reviewers for their time and insightful and constructive comments. After the first round of revision, we are pleased that reviewers found “*the results are improved*” and conclusions are “*more solidified and are significant findings for the gene regulation field*”. We have addressed the points raised by the reviewers and revised the manuscript by reanalyzing certain genomic data sets and clarifying writing as described below. Changes in the manuscript are underlined.

Reviewer #2 (Remarks to the Author):

Remarks to the Author Much of the data appears to have been reanalyzed and the results are improved. The observations of regulation through pause release, transient and P-TEFb independent dismissal of NELF, and NELF-dependent attenuation of the transcription factors that activate Il10, are more solidified and are significant findings for the gene regulation field.

Major comments remaining are below:

As mentioned by another reviewer, Pausing is widespread in most cells types examined and highly correlated with NELF occupancy. Thus the statement in the abstract, “Here, we report widespread promoter-proximal RNA polymerase II pausing in resting macrophages, marked by broad co-localization of the negative elongation factor (NELF) complex. . .” could be said about virtually any cell type and is not specific to macrophages. They really should back off from this as a significant claim in the abstract, results, and discussion.

Pol II promoter-proximal pausing is conserved from *Drosophila* to mammals. Previous studies have showed that about 30% of all genes display paused Pol II in human primary lung fibroblasts, mouse ES cells or *Drosophila* cells (Adelman K. *et al*, Nature Reviews Genetics 2012, cited in the manuscript). The fraction of genes harboring paused Pol II appears to be relatively constant across species and developmental stages. We agree that in this sense, our data in resting macrophages are largely consistent with the expected patterns. We have modified text in results and discussion sections as suggested (pages 6 and 14).

It’s pretty clear from the heatmap in figure 1d, that group 2 and 3 might be called partially as a result of the annotation list used. As pointed out by reviewer #3, using the longest annotation is not ideal. In most cases, you will end up using the most 5’-isoform of the gene. Since the TSS is then defined as +/-250 bp from the TSS, genes with more active isoforms downstream of this window will be miscategorized.

It can be clearly seen in the heatmap as the apparent peak right downstream of the TSS window. Thus, many of the group 3 genes should be in group 1 or 2 if the correct TSS was used. In the end, it might not affect the results and conclusions here, but it does look like a sloppy analysis oversight in the figure.

We agree that accurately depicting TSS positions will result in more precisely defined pausing status for each gene. In line with this notion, we analyzed the number of annotated TSS for each gene among total unique genes (n=24,030) used in our dataset and found that only 8% of genes (n=1,973) showed multiple TSS. Therefore, although our analysis using longest annotated variant may not be ideal for all genes, this methodology indeed recapitulated the pausing status for the vast majority of genes in the genome.

Methods on the chromatin fractionation for detecting NELF association should be included.

The methods for detecting NELF association on the chromatin fractionation are now described on page 20. In brief, the low-salt extracted nuclei were collected as the chromatin-associated fraction. Then, NELF association was detected by immunoblotting using antibodies for NELF-B (EPR11200) (Abcam, ab167401) and NELF-E (Proteintech, 10705-1-AP).

The conclusion that PU.1 “facilitates Pol II pausing” is still misleading (pointed out by reviewer #3 earlier). PU.1 regulation of genes could actually be through recruitment and initiation of the transcription machinery, making a role for pausing indirect. As stated, the sentence implies that PU.1 is involved in pausing. This cannot be concluded from the current data. This section is actually a distraction from the rest of the work and the authors should consider removing it.

According to our data, PU.1 binding around the TSS regions contributed to the TSS-centric Pol II and NELF occupancy in macrophages, yet, we agree with the reviewer that these results did not exclude a plausible role for PU.1 in initiation. We have modified the title and wording throughout the section to soften the claim (pages 6-7). Actually, this line of investigation represents the first attempt to explore the role

of TSS-centric PU.1 in macrophages and thus, we believe, that it would be beneficial for these results to be released and to potentially spur future studies on this topic.

Have the authors generated PRO-seq in the 0.5 hours after LPS treatment? Since less NELF is associated to TSS, it would be interesting to see if you get an expected (at least in this case) increase pause release above what is seen at 1 hour.

Our PRO-seq data were generated with WT and *Nelfb* KO BMDM treated with or without LPS for 0.5 h (described on page 18-19). In order to maximize data consistency, the current PRO-seq data sets were generated from the same batch of primary macrophages from each genotype to ensure that differences among samples at different time points were not due to intrinsic differences in cells. In order to add one additional time point, we would have to repeat the entire PRO-seq experiment with WT and *Nelfb* KO macrophages, including untreated controls, which is a tremendous undertaking (~ 50×10⁶ cells per condition). We believe that the extensive genomic analyses supported the conclusions reasonably well and that additional data may not yield sufficient new insight to justify the effort.

Reviewer #3 (Remarks to the Author):

A previous point that I do not believe was adequately addressed was the fact that 3/4 of the genome BMDM transcriptome is paused. I don't believe that this is different than any other cell type, so I am unsure why this is highlighted as being substantial or meaningful.

We agree with the reviewer that Pol II promoter-proximal pausing is conserved from *Drosophila* to mammals and the fraction of genes that display Pol II pausing appears to be relatively constant across species and developmental stages. We have modified writing to clarify that this is not a macrophage-specific phenomenon (page 6).

I am not in favor of the authors using and reporting p-values that are not multiple testing corrected in their analysis. While they do multiple testing correct in the RNA-seq by thresholding on q-value, they do not for other tests. I was unable to assess the appropriate nature of the statistics in the previous version because an adequate methods section was not provided.

We used non-adjusted p-value to identify differentially expressed genes between NELF KO and WT cells to be more inclusive. Indeed, using multiple testing corrected p-value (q-value) results in relatively few genes identified as differentially expressed, making it difficult to evaluate the significance of NELF-regulated genes during inflammation. Importantly, we performed qPCR to independently validate expression patterns of functionally important genes with great reproducibility (Figures 5 and 6). Thus, our less stringent statistical criteria in RNA-seq gene selection did not hamper overall data interpretation.

The authors use TTS (transcription termination site), when they intend to use "cleavage and poly-Adenylation site"

We used the same nomenclature as the UCSC annotation according to which the TTS (transcription termination site) refers to the cleavage and poly-adenylation site, as the annotation of each gene is derived from the sequence of mature poly-adenylated mRNA. We have now revised TTS description to reflect this point (pages 23-24).

REVIEWERS' COMMENTS:

Reviewer #2 (Remarks to the Author):

The response from the authors has sufficiently addressed my concerns.